# Pseudo Rotary Resonance Relaxation Dispersion Effects in Isotropic Samples

**Authors:** *Evgeny Nimerovsky\*, Jonas Mehrens & Loren B. Andreas\**

**Affiliations:**

Department of NMR based Structural Biology, Max Planck Institute for Multidisciplinary

Sciences, Am Faßberg 11, Göttingen, Germany

*Corresponding authors: land@mpinat.mpg.de ORCID: 0000-0003-3216-9065 and

evni@mpinat.mpg.de ORCID: 0000-0003-3002-0718.

**Abstract**

Enhanced transverse relaxation near rotary-resonance conditions is a well-documented effect for anisotropic solid samples undergoing magic-angle spinning (MAS). We report transverse signal decay associated with rotary-resonance conditions for rotating liquids, a surprising observation, since first-order anisotropic interactions are averaged at a much faster timescale as compared with the spinning frequency. We report measurements of $^{13}$C and $^{1}$H signal intensities under spin-lock for spinning samples of polybutadiene rubber, polyethylene glycol solution and 99.96% $D_2O$. A drastic reduction in spin-lock signal intensities is observed when the spin-lock frequency matches one or two times the MAS rate. In addition, oscillations of the signal are observed, consistent with a coherent origin of the effect, a pseudo rotary-resonance relaxation-dispersion (pseudo-RRD). Through simulations, we qualitatively describe the appearance of pseudo-RRD, which can be explained by time dependence caused by sample rotation and an

inhomogeneous field, the origin of which is an instrumental imperfection. Consideration of this
effect is important for MAS experiments based on rotary-resonance conditions, and motivates the
design of new MAS coils with improved rf-field homogeneity.
**KEYWORDS:** Magic-angle spinning, nuclear magnetic resonance spectroscopy, pseudo rotary-
resonance relaxation-dispersion effect
**Introduction**

7       Measurement of the transverse relaxation rates of nuclear spins as a function of the

applied rf-field spin-lock strengths is an elegant and well-established method for detecting
structural molecular dynamics.(Abyzov et al., 2022; Alam et al., 2024; Camacho-Zarco et al.,
2022; Hu et al., 2021; Massi and Peng, 2018; Palmer, 2015; Palmer and Massi, 2006; Pratihar et
al., 2016; Rangadurai et al., 2019; Sekhar and Kay, 2019; Stief et al., 2024) For molecular solids,
rocking motion or slow exchange in organic and inorganic samples(Fonseca et al., 2022; Keeler
and McDermott, 2022; Krushelnitsky et al., 2018, 2023; Kurauskas et al., 2017; Lewandowski et
al., 2011; Ma et al., 2014; Marion et al., 2019; Öster et al., 2019; Quinn and McDermott, 2009;
Rovó and Linser, 2018; Shcherbakov et al., 2023; Vugmeyster et al., 2023) under MAS(Andrew
et al., 1958; Lowe, 1959) NMR have been studied via the impact on transverse relaxation. This
detection can be achieved by performing a spin-lock experiment,(Furman et al., 1998) where the
decay of magnetization is measured as a function of the power of the applied spin-lock (SL)
pulse. For slow motion or slow exchange in the microsecond (µs) range, the spectral
densities(Redfield, 1957) of the investigated spins may include additional terms(Kurbanov et al.,
2011; Marion et al., 2019) that arise from non-averaged anisotropic interactions.(Kurbanov et al.,
2011; Rovó, 2020; Schanda and Ernst, 2016) These terms depend on the sums and differences
between the nutation frequency induced by the rf-field ($\nu_{SL}=\gamma B_1/(2\pi)$) and MAS rate ($\nu_R$). Such
dependence causes a significant increase in the measured relaxation rates when $\nu_{SL}$ approaches
one of the rotary-resonance conditions ($\nu_{SL} = \nu_R$ or $2\nu_R$).(Marion et al., 2019)
For liquid samples, where SL experiments are routinely used to detect fast exchange,(Cavanagh
et al., 2006; Deverell et al., 1970; Palmer, 2004) sample rotation is not expected to induce any
rotary-resonance conditions based on anisotropic spin interactions,(Levitt et al., 1988; Oas et al.,
1988) since such interactions are eliminated by nanosecond-timescale isotropic
motion.(Haeberlen and Waugh, 1968; Maricq, 1982) However, to our surprise, we still observed
changes in the SL signals at rotary-resonance conditions for liquid and liquid-like samples during
SL experiments. Since the signal decreases, but is also clearly oscillatory, a signature of coherent
effects, we refer to this phenomenon as a pseudo rotary-resonance relaxation-dispersion (pseudo-
RRD). A review of the literature revealed articles suggesting related resonance conditions for
rotating liquid samples: in adiabatic TOCSY experiments, enhanced performance was observed
under specific matching conditions in relation to the spinning frequency.(Kupče et al., 2001;
Zektzer et al., 2005)
In this article, we measured pseudo-RRD for several liquid and liquid-like samples, and observe
similar effects in each. Through numerical simulations,(Nimerovsky and Goldbourt, 2012) we
show that this behavior can be qualitatively explained by the influence of the periodic component
of the applied rf-field, which arises from the rotation of the sample in a spatially inhomogeneous
rf-field.(Aebischer et al., 2021; Tošner et al., 2017)
**Results and Discussion**
We measured pseudo RRD for natural abundance $^{13}$C polybutadiene rubber at 10 kHz, 20
kHz and 35 kHz MAS. The same pseudo RRD behavior is observed for a polyethylene glycol
solution at 10 kHz MAS and for residual protons in liquid deuterium oxide (99.96%). The
polybutadiene rubber displays liquid-like spectra but does not undergo translational diffusion due
to the elastomeric properties of a cross-linked polymer. On the other hand, since the
polybutadiene is an elastomer and therefore may not undergo perfect isotropic averaging, we also
recorded data for a polyethylene glycol solution and liquid water.

6         Figure 1 displays the spin-lock sequence. Similar to previously proposed

versions,(Vugmeyster et al., 2022) it contains a heat compensation block(Wang and Bax, 1993)
(HC), followed by a $\pi/2$-pulse, $T_2$ −filter(Schmidt-Rohr et al., 1992) (to reduce any broad signal
components from the polymer) and a spin-lock pulse (SL). The mixing times for HC and SL
pulses were the same during a single experiment ($t_{HC} = t_{SL} = N_{SL}T_R$), while the sum of the rf-
field powers of these applied pulses always equaled to a fixed value. In all experiments, we used
continuous HC and SL (Figure 1B) except in one (the data is shown in Figure 2C), where we
applied windowed pulses (Figure 1B). During acquisition, WALTZ16 decoupling(Shaka et al.,
1983) was used.

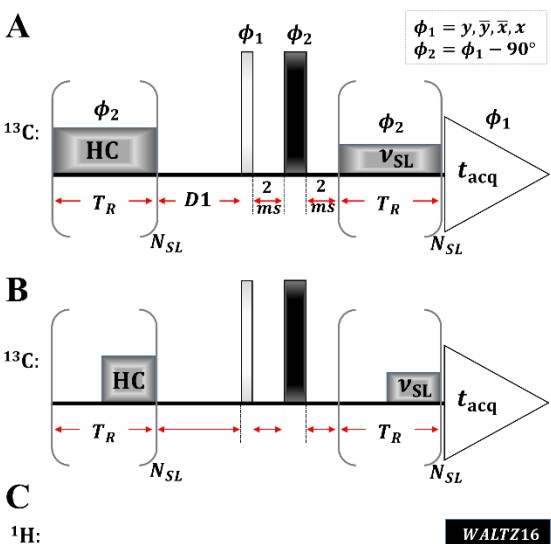

**Figure 1** Spin-lock sequence with heat compensation (HC), $T_2$ −filter (2 ms – $\pi$-pulse – 2ms) and spin-lock (SL)
blocks. The SL and HC elements consisted of a train of $N_{SL}$ rotor-synchronized continuous (A) or windowed (B)
pulses with the same phase ($\phi_2$) and rf-field strength ($\nu_{SL}$). In all experiments, $power_{HC} + power_{SL}$ =constant
(equivalent to 50 kHz rf-field strength). During acquisition, WALTZ-16 decoupling(Shaka et al., 1983) (C) was
applied on the $^1$H channel.

6         The experimental $^{13}$C polybutadiene rubber SL profiles (acquired with a 1.3 mm probe)

under three different MAS rates: 10 kHz (A and C), 20 kHz (D) and 35 kHz (B) are shown in
Figure 2. For Figures 2A, 2B and 2D, a drastic reduction in the SL signal is observed at rotary-
resonance conditions when $\nu_{SL}$ equals either $\nu_R$ or $2\nu_R$. Together with reduction in the SL signal,
oscillations are observed. For Figure 2C, we used 10 kHz MAS and windowed pulses: half of the
rotor period is a window, as shown in Figure 1B. Again, a drastic reduction in the SL signal is
observed, but when $\nu_{SL}$ equals either to $2\nu_R$ or $4\nu_R$. We previously observed similar behavior for
windowed CP profiles,(Nimerovsky et al., 2023) where increasing the window between rotor-
synchronized pulses from zero to half a rotor period doubled the required rf-field strength for
cross-polarization transfers.(Hartmann and Hahn, 1962) Interestingly, with windowed pulses, the
SL profile appears similar to that with continuous pulses, and even under a low rf-field strength
of 1 kHz, there is no change in the SL signal intensities (Figure S1A in supplementary
information, SI). The experimental spin-echo(Hahn, 1950) and inversion recovery(Vold et al.,
1968) curves for this sample are illustrated in Figure S1A-B in SI.
From Figure 2, we can also observe that the location of the first minimum signal intensity in the
experimental SL profiles depends on the MAS rate (indicated in gray in Figure 2). For 10 kHz
MAS (Figure 2A and 2C), the locations are approximately at a 3 ms SL time, while for 20 kHz
(Figure 2D) and 35 kHz (Figure 2B), the locations are approximately at 1 ms and 0.4 ms,
respectively. However, in all four profiles at these minimum points, the signal reaches a similar
value of approximately 0.53.

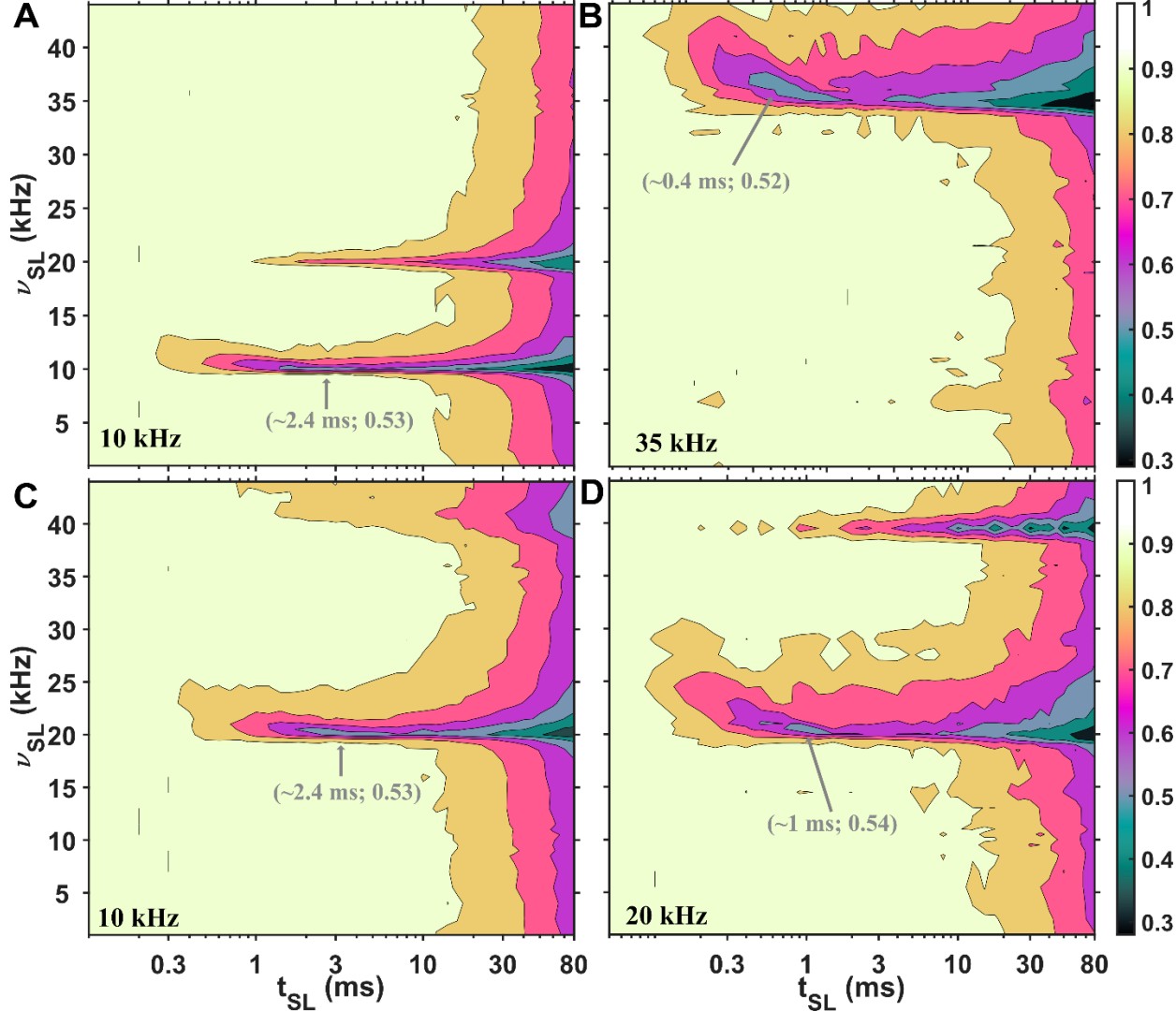

**Figure 2** $^{13}$C polybutadiene rubber signal (the peak intensities) is shown as functions of the rf-field strength ($\nu_{SL}$, y-
axis) and mixing time ($t_{SL}$, x-axis) of the SL under three different MAS rates: 10 kHz (A and C), 20 kHz (D) and 35
kHz (B). For (A), (B) and (D), continuous SL was applied, while for (C), windowed (half rotor period was filled
with the pulse) SL was implemented. The values in gray represent the coordinates of the first minimum in the
profiles. Additional experimental details are provided in the supplementary information (SI).

1        Rotary-resonance conditions at $\nu_R$ and $2\nu_R$ of rf-field strength are also observed for the

2        polyethylene glycol (Figure 3B, acquired with a 4 mm probe) and for residual protons in liquid

3        deuterium oxide samples (Figure 3D, acquired with a 1.3 mm probe). The 1D spectra of these

4        samples are shown in Figure 3A and C, for PEG and liquid water. For each sample, two rotary-

5        resonance conditions are observed at positions equal to integer multiplies of the MAS rates

6        $(\nu_{SL} = \nu_R, 2\nu_R)$. For liquid water (Figure 3D), the additional rotary-resonance condition with n=3

7        appears very weakly. We more carefully sampled around this condition for the water sample.

8        The performance of the SL experiments on all three samples helps rule out the influence of

9        translational diffusion(Hahn, 1950) (which may be present for polyethylene glycol and liquid

10       water but not for polybutadiene rubber) or residual dipolar interaction(Cohen-Addad and Vogin,

11       1974) (which might be present for polybutadiene rubber but is not relevant for polyethylene

12       glycol and liquid water).

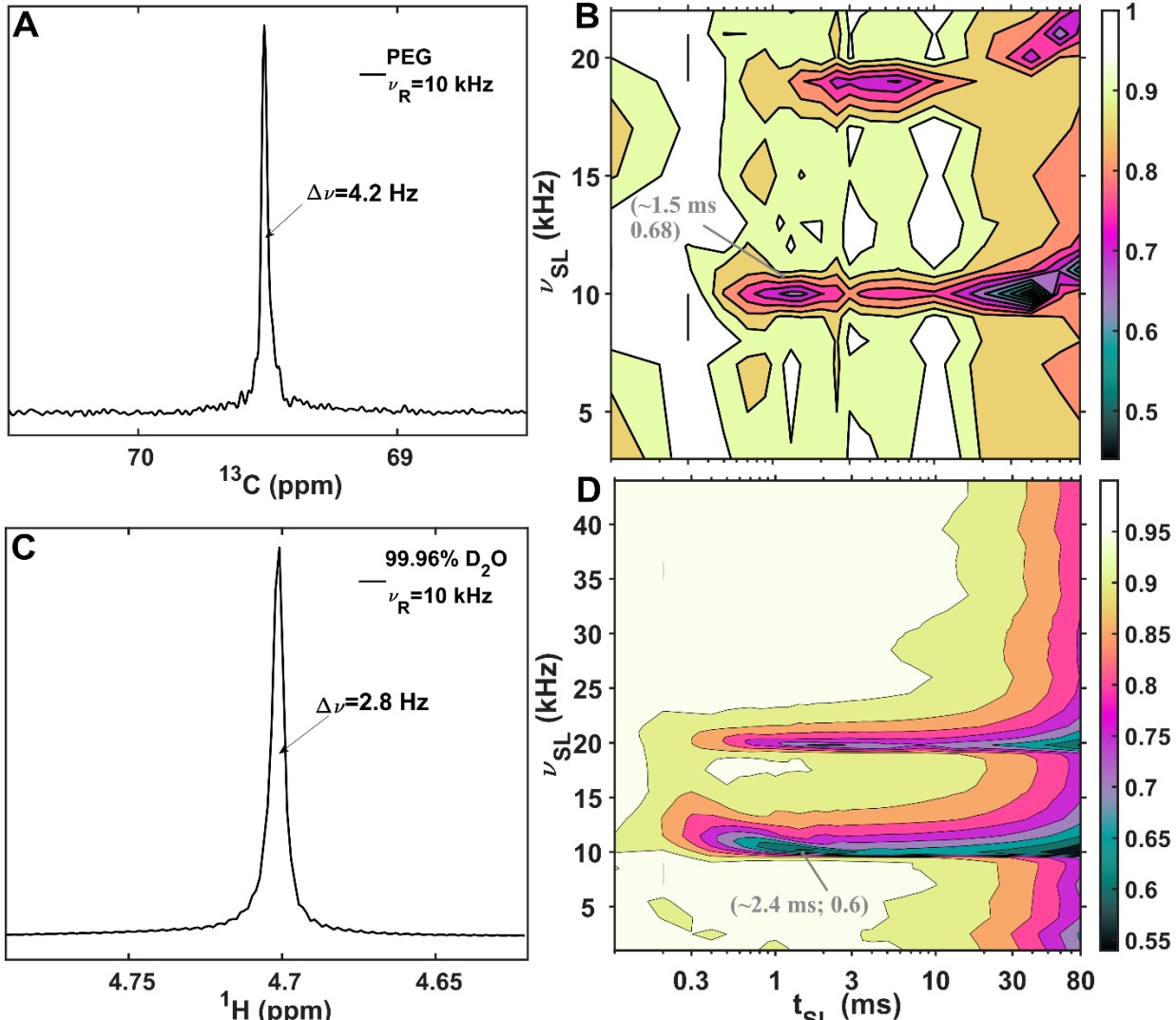

**Figure 3** $^{13}C$ and $^{1}H$ spin-lock profiles at 10 kHz MAS. (A-B) Single-pulse $^{13}C$ spectra and SL profile of polyethylene glycol (PEG) acquired with a 4 mm probe. (C,D) $^{1}H$ single pulse and SL profile of 99.96% $D_2O$ acquired with a 1.3 mm probe. The profiles in (B,D) show $^{13}C$ and $^{1}H$ signal amplitudes (peak intensities) as a function of the rf-field strength ($\nu_{SL}$, y-axis) and mixing time ($t_{SL}$, x-axis) of the SL pulse. The values in gray are the coordinates of the first minimum in the profiles. Additional experimental details are provided in the supplementary information (SI).

To understand the major source of the apparent rotary-resonance conditions in liquid and liquid-like samples, we performed numerical simulations. In these simulations, two scenarios

were considered in which the external magnetic fields ($B_0$ or $B_1$) gained time dependence due to
the rotation of the sample and:
- A spatially inhomogeneous external magnetic field strength ($B_0$ modulation),
- A spatially inhomogeneous rf-field strength ($B_1$ modulation).
Note that the first scenario is compatible with a narrow linewidth under MAS.(Sodickson and
Cory, 1997) The specific spatial distributions chosen for $B_0$ and $B_1$, which, along with MAS,
introduce modulations, are summarized in Table S1 and shown in Figure S7 in SI. Additionally,
for each scenario, a time-independent distribution of rf-field strength was also included. Its
profile and influence has been investigated previously(Engelke, 2002; Gupta et al., 2015; Hoult,
1976; Paulson et al., 2004; Tošner et al., 2017, 2018). This type of inhomogeneity is expected for
solenoidal coils in which the sample near the ends of the coil experiences a lower rf-field
strength. For simplicity, we did not include a time-independent distribution of external magnetic
field strength,(Hoult, 1976; Hürlimann and Griffin, 2000) as it had a minor effect in our
experiments, and shimming under MAS ensures that it is minimal.
The effects of inhomogeneous rf-field on MAS spectra have been investigated
previously.(Aebischer et al., 2021; Goldman and Tekely, 2001; Tekely and Goldman, 2001;
Tošner et al., 2017) Rather than $B_1$ oscillations, the coil receptivity was shown to oscillate due to
rotation of the sample relative to the coil, and the authors showed that this instrumental
imperfection results in the appearance of sidebands that are unrelated to the chemical shift
anisotropy (CSA).( Goldman and Tekely, 2001; Tekely and Goldman, 2001) Sidebands due to
rotation through inhomogeneous $B_0$ and $B_1$ fields is a well-known effect in liquids.(Malinowski
and Pierpaoli, 1969; Vera and Grutzner, 1986) For solid samples, Aebischer et al.(Aebischer et
al., 2021) investigated the influence of time-dependent modulations of the rf-field amplitude and

phase on the performance of selected recoupling sequences and nutation experiments. In this case, the modulations did not significantly affect most recoupling sequences, with the exception of double quantum C-symmetry sequences.(Lee et al., 1995) It was noted much earlier that oscillations in phase were needed to fully explain experimental results in rotary resonance recoupling.(Levitt et al., 1988) Consistent with the matching conditions identified in this study, Aebisher et al.(Aebischer et al., 2021) revealed significant effects at $\nu_R$ and $2\nu_R$ in nutation spectra. The distribution of $B_1$ fields in a solenoidal coil was elegantly visualized in SL experiments of solid samples, in which case the loss of signal at rotary resonance was interpreted as CSA recoupling.(Tošner et al., 2017)

In SI, using average Hamiltonian theory(Haeberlen and Waugh, 1968) and numerical simulations, we show that the appearance of rotary-resonance conditions is primarily related to the presence of an additional time-dependent periodic term in the Hamiltonian, orthogonal to the applied RF-field spin-lock (Eqn. S1 in SI). Starting with simulations using single amplitude modulations (Figures S3-S5 in SI), we extend them to cases where these modulations have spatial distributions (Figure S6 in SI and Figures 4-5). For the qualitative determination of the source of the pseudo-RRD, we use Gaussian and bi-quadrate exponential distributions for the time-dependent and time-independent $B_1$ modulations, respectively (summarized in Table S1 and shown in Figure S7 in SI). While knowing the exact values and shapes of these modulations is important for quantitatively assessing their influence, these values and shapes primarily broaden the rotary-resonance conditions and affect the frequency and amplitude of the modulations in the spin-lock signals. However, they do not affect the occurrence of the rotary-resonance conditions. Consequently, rotary-resonance conditions are observed at the same MAS frequencies for both single (Figure S5 in SI) and distributed (Figure S6 in SI) $B_1$ modulations.

Figure 4 shows simulations for the first scenario ($B_0$ modulation). While some similarities
between Figure 2 and Figure 4 are observed, there are three major differences in the SL profiles,
which should be highlighted. Firstly, in Figure 4, the intensities at the first minima show a
dependence on MAS rate (marked in gray in Figure 4), whereas in Figure 2, the experimental
profiles show only a slight dependence. Secondly, in Figure 4, the locations of these minima in
time (x-axis) do not depend on the MAS rate (Figure 4A, 4B and 4D), whereas the location in
time changes when windowed pulses are applied (Figure 4C). In contrast, the experimental
profiles exhibit the reverse behavior. Thirdly, in Figure 4C, second rotary-resonance condition is
slightly observed, compared to Figures 4A and D, while in Figure 2C, two rotary-resonance
conditions are clearly detected. Additionally, increasing the magnetic field inhomogeneity by
deliberately mis-setting the room temperature shims had little influence on the SL profile (shown
in Figure S2 in the SI).
All of this indicates that a spatially inhomogeneous external magnetic field cannot be a major
source of the appearance of rotary-resonances conditions in rotating liquids and liquid-like
samples.

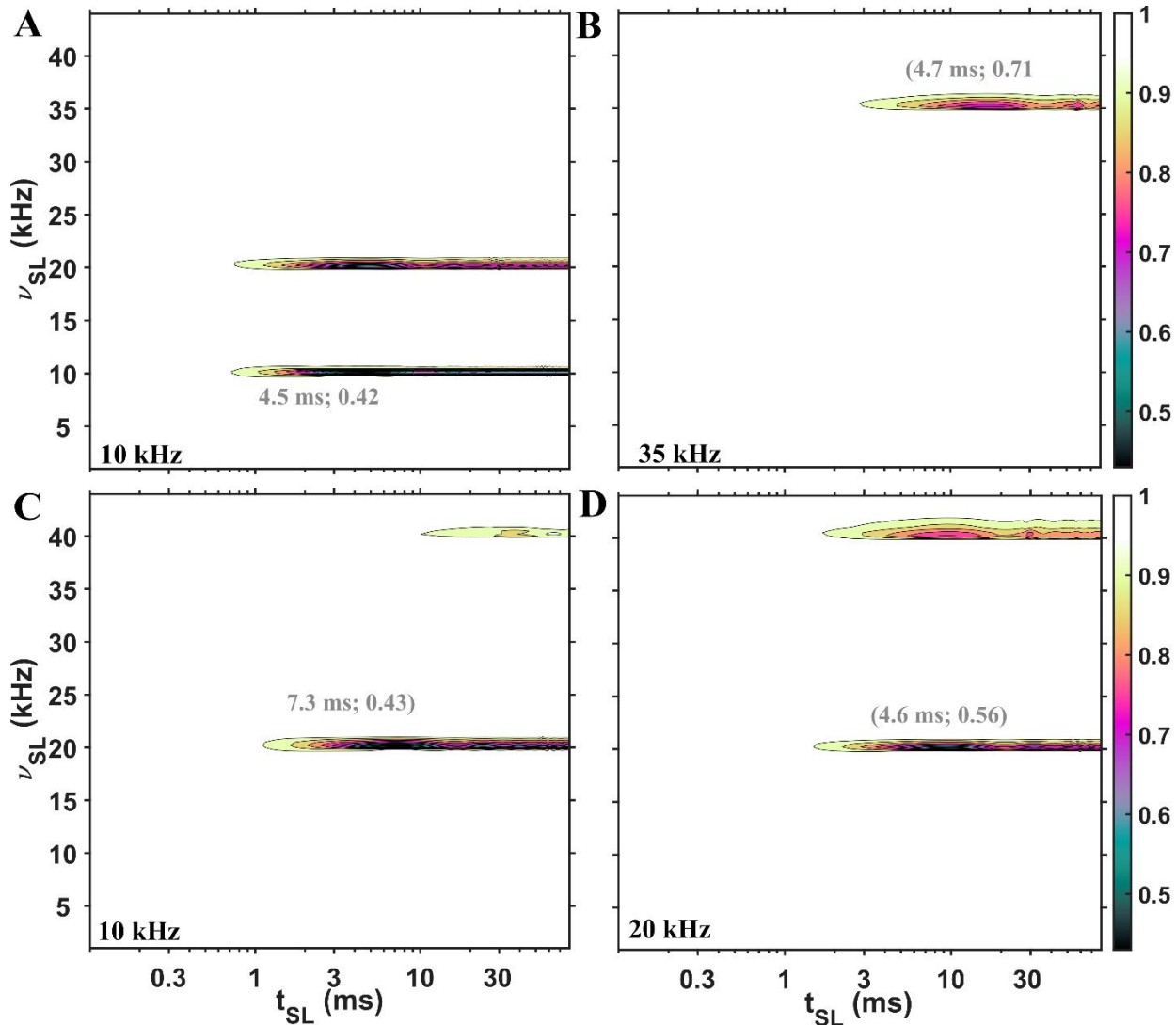

**Figure 4** Simulated SL profiles showing the influence of time dependence introduced via an inhomogeneous

external magnetic field. The simulated signal is shown as a function of the rf-field strength ($v_{SL}$, axis y) and mixing

time ($t_{SL}$, axis x) of the SL under three different MAS rates: 10 kHz (A and C), 20 kHz (D) and 35 kHz (B). For (A),

(B) and (D), continuous SL was applied, while for (C) windowed SL was implemented (half rotor period was filled

with the pulse). The values in gray represent the coordinates of the first minima in the profiles. No

phenomenological relaxation was included in the simulations. Additional simulated details are provided in the SI.

In contrast, simulations of SL profiles with time dependence introduced via a spatially

inhomogeneous rf-field (Figure 5, B₁ modulation) qualitatively agree with the experimental

plots, indicating that a spatially inhomogeneous rf-field is a better explanation for the appearance

of rotary-resonance conditions in rotating liquids and liquid-like samples using conventional MAS NMR probes with solenoidal coils. Such time dependence has been previously considered in the design of magnetization transfer elements using optimal control.(Blahut et al., 2022, 2023; Glaser et al., 2015; Joseph and Griesinger, 2023; Tošner et al., 2017, 2018)

This qualitative explanation, provided by simulations, indicates that this effect can also be anticipated in experiments involving solid samples, in addition to the desired effects caused by molecular motion. It is therefore recommended to consider coil inhomogeneity when measuring relaxation rates near rotary resonance conditions. Fortunately, the magnitude of this effect is considerably smaller than the strong relaxation observed in recent reports that detected slow structural dynamics via near rotary resonance conditions.(Krushelnitsky et al., 2018)

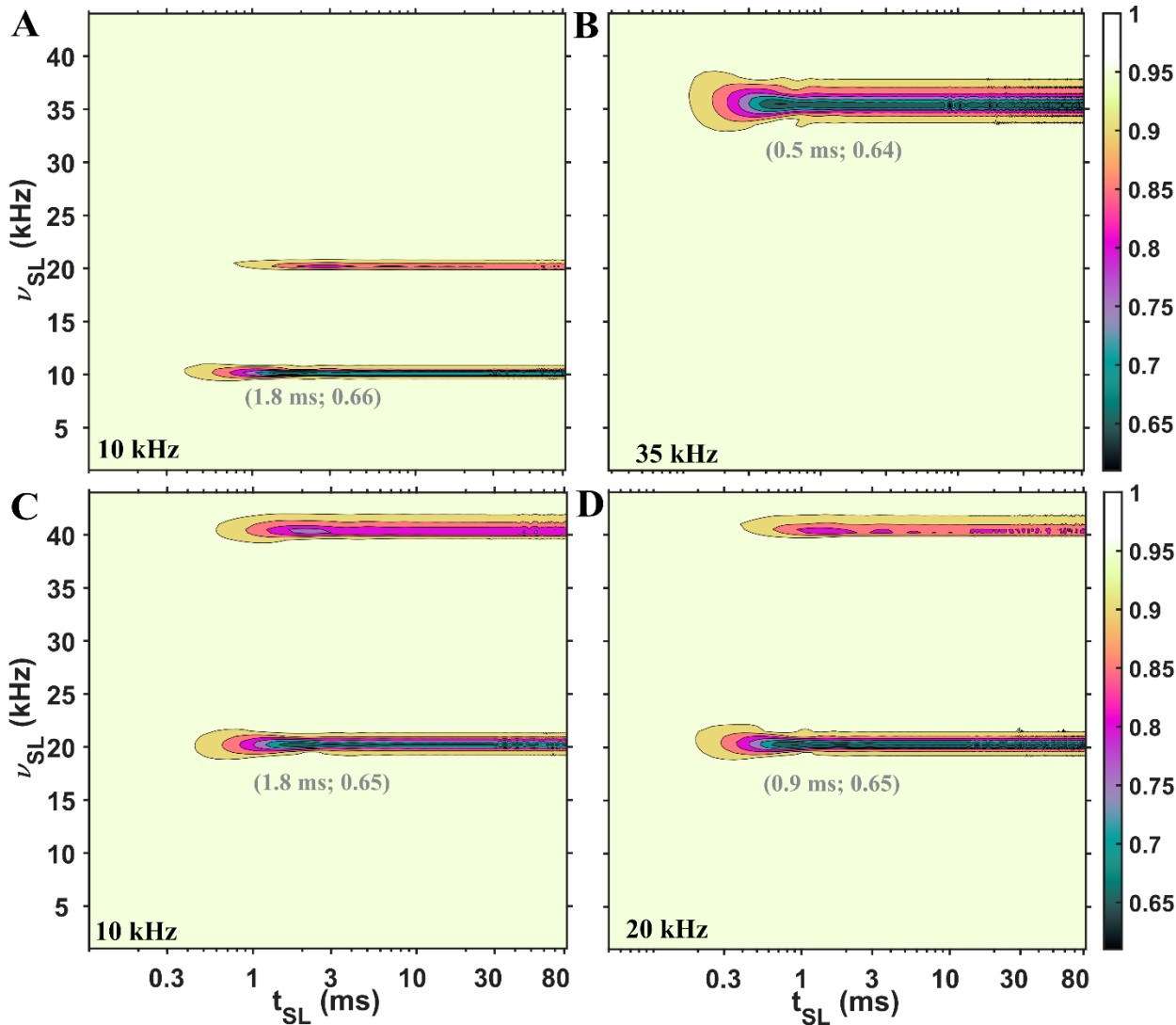

**Figure 5** The influence of the inhomogeneous rf-field on the simulated SL profiles. The simulated signal is shown as functions of the rf-field strength ($\nu_{SL}$, axis y) and mixing time ($t_{SL}$, axis x) of the SL under three different MAS rates: 10 kHz (A and C), 20 kHz (D) and 35 kHz (B). For (A), (B) and (D), continuous SL was applied, while for (C) windowed (half rotor period was filled with the pulse) SL was implemented. The values in gray represent the coordinates of the first minima points in the profiles. Relaxation was not included in the simulations. Additional simulated details are provided in the SI.

**Conclusions**

Rotary-resonance conditions, under which the applied rf-field strength equals an even multiple of the MAS rate, provide a powerful avenue to obtain specific structural information via recoupling of anisotropic interactions in solids(De Paëpe, 2012; Nishiyama et al., 2022) or for detecting changes in the relaxation rates due to slow motion in the μs range.(Rovó, 2020) Canonically, rotary-resonance conditions are not expected in liquids due to the averaging of first-order anisotropic interactions from (sub) nanosecond isotropic motion.(Haeberlen and Waugh, 1968; Maricq, 1982) In this article, we presented experimental data, in which we detected rotary-resonance conditions in a liquid and a liquid-like samples. We qualitatively explained the major source of these conditions, which can occur from a combination of two factors: the rotation of the sample and a spatially inhomogeneous rf-field produced a solenoidal coil.(Tošner et al., 2017) As a result, the rf-field Hamiltonian contains time-dependent terms, which leads to signal decrease, i.e. pseudo relaxation behavior, at or near rotary-resonance conditions. To mitigate these effects, it may be advantageous to consider different hardware designs,(Chen et al., 2018; Xu et al., 2021) for example rf coils that produce more homogeneous rf-fields.(Grant et al., 2010; Kelz et al., 2019; Krahn et al., 2008; Stringer et al., 2005)

**Competing interests**

The contact author has declared that none of the authors has any competing interests.

**Acknowledgments**

We acknowledge financial support from the MPI for Multidisciplinary Sciences, and from the Deutsche Forschungsgemeinschaft (Emmy Noether program Grant AN1316/1-2). We thank Dr. Supriya Pratihar for inspiring this work by noticing pseudo-RRD at rotary resonance conditions in exploratory high-power relaxation dispersion measurements in a 0.7 mm MAS probe. We thank Dr. Dirk Bockelmann and Brigitta Angerstein for technical assistance.

## Author contribution

EN and LBA designed the experiments. EN and JM recorded NMR data and ran simulations, EN and LBA wrote the article. All authors edited and approved the article.

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
