# Peer review of "Pseudo Rotary Resonance Relaxation Dispersion Effects in"

_Magnetic Resonance, 2024_

## Referee Comment (RC1)

Pseudo Rotary Resonance Relaxation Dispersion Effects in Isotropic Samples

E. Nimerovsky, J. Mehrens, L. B. Andreas

In this manuscript, the authors report the experimental observation of rapid signal loss during a spinlock pulse in MAS NMR experiments of isotropic samples when irradiating at a rotary resonance condition (nu_1 = n · nu_R, with n = 1, 2). This phenomenon is well-known for solid samples, where irradiating at one of the rotary resonance conditions recouples the chemical-shift anisotropy as well as the anisotropic heteronuclear dipolar coupling which leads to coherent broadening and, therefore, signal decay. Moreover, irradiating at the rotary resonance conditions leads to a sampling of the spectral-density function at zero and can, thus, lead to enhanced rotating-frame relaxation (T1rho, incoherent broadening) in dynamic systems. In this work, however, the authors report enhanced signal decay in MAS experiments of polybutadiene rubber and polyethylene glycol solution samples that exhibit liquid-like spectra and are, thus, considered isotropic. They attribute the observed signal decay at the rotary resonance conditions to the MAS modulation of the applied rf field due to the spatial inhomogeneity of the rf field. Numerical simulations of spinlock experiments that are performed using a relatively basic model of the spatial rf-field distribution generated by a solenoid coil appear to support this explanation.

Although the work reports a potentially relevant phenomenon, there are notable issues that need to be addressed. My major concern is that the simulation results presented in the paper are not reproducible based on the description provided in the SI. The conclusions drawn by the authors about the enhanced signal decay originating from the MAS modulation of the applied rf field solely based on the simulation results that are presented therefore appear to be unsubstantiated. Additionally, the model employed for the spatial distribution of the RF field generated by a solenoid coil in a typical MAS NMR probe is overly simplistic and may overestimate the magnitude of phase errors arising from the radial rf-field inhomogeneity. This raises questions about the validity of the authors' interpretation of their simulation results. To enable a proper assessment of the manuscript and determine whether it can be recommended for publication, it is essential that the authors provide additional information regarding their numerical simulations and address the specific concerns outlined below.

1)  Numerical simulations

The description of the numerical simulations provided in the SI suggests that the authors performed simulations for a single-spin-1/2 system by numerical integration of the Liouville-von Neumann equation. Based on Eq. (S2), both the initial density operator as well as the detection operator correspond to Ix (x magnetization). For simulations taking the spatial rf-field inhomogeneity into account, the Hamiltonian of the system corresponds to a spinlock along the x-axis (Eq. (S3C)) and an orthogonal component along the y-axis that is modulated by the MAS frequency (Eq. (S3B)). This will generate an effective field whose orientation and magnitude will be modulated by the MAS frequency and about which the magnetization will nutate. Due to spatial inhomogeneity of the rf field, the orientation and magnitude of this effective field will vary across the sample space. The authors model the spatial rf-field distribution using distribution functions f_B1 (that models the amplitude distribution along the rotor axis) and f_nu1 (that models the

magnitude of the modulated component along the y-axis) and compute the total signal as the weighted average of these distributions (Eq. (S2)). Unfortunately, the functional forms of these distribution functions are not provided (which they should be) and can only be approximated based on Fig. S1. Since no other spin-spin or spin-field interactions are mentioned, it must be assumed that no other interactions are taken into account, which would be in line with an isotropic sample of an isolated nucleus. Moreover, the authors explicitly say that no relaxation is taken into account in the simulations and the effect therefore has to be coherent in nature. However, computing the signal based on this information (which is computationally inexpensive for a single-spin system) does not lead to the same results presented by the authors in Fig. 3 in the main text. I would therefore ask the authors to provide additional information on how the simulations were performed.

2) Model for the rf inhomogeneity

In my opinion, the model used for the spatial distribution of the rf-field in a solenoid coil is not appropriate due to the following issues:

- The authors assume an amplitude deviation from the nominal spinlock amplitude of only 0-5% (Eq. (S3C)). However, both simulated (e.g. Tosner et. al. 2017) and measured (e.g. Gupta et. al. 2015) rf-field distributions of common MAS NMR probes suggest that the field drops to approximately 50% at the edges of the sample space. Since there is no mention of spatial sample restriction to the central part of the rotor, the chosen amplitude distribution is not appropriate.
- For a maximum deviation of the spinlock amplitude of 5% the signal weighting function given in Fig. S1C is not appropriate. According to the reciprocity theorem (e.g. Hoult 1976), the induced signal is directly related to the B1 field of the receiving coil at a given position in the sample space. A deviation from the nominal field by 5% should therefore not lead to a weighing of the signal with a factor of close to zero as is shown in Fig. S1C.
- The orthogonal component (Eq. (S3B)) occurs due to the MAS modulation of the phase of the rf irradiation which is known to be strongest at the edges of the sample space (e.g. Tosner 2017). Since the spinlock amplitude is also significantly lower than the nominal spinlock amplitude in these parts of the sample, the rotary resonance condition is no longer fulfilled in these parts of the sample and the coil sensitivity is significantly lower. However, the model implemented by the authors uses the same magnitude of the modulation for all values of f_B1.

Since simulated realistic rf-field distributions can be found in the literature (Tosner et. al. 2017), I would suggest that the authors perform their simulations using such a distribution instead of this crude model that appears to overestimate the magnitude of the phase errors due to the rf-field inhomogeneity.

3) Isotropic sample

The experimental results presented in the paper stem from measurements of a polybutadiene rubber sample as well as a polyethylene glycol sample. Although these sample exhibit "liquid-like" spectra, they both correspond to large polymer systems, where partial alignment might lead to residual anisotropic dipolar interactions that will be recoupled for irradiation at a rotary resonance condition. The comparison of experimental spectra of the rubber sample with and

without MAS (Fig. S4A) shows that the static spectrum is significantly broader than the one under MAS. This line broadening can either be attributed to residual anisotropic interactions or susceptibility effects and raises the question if either of the two are re-introduced by irradiating at the rotary resonance condition which would lead to signal decay.

The authors attribute the observed signal decay solely to the modulation of the rf-field due to the rf inhomogeneity, based on numerical simulations of a single-spin system. This raises the question why the authors didn't choose to demonstrate the effect using a truly isotropic system (such as, for example, the residual H2O line in D2O) that would correspond more closely to what is simulated.

4) Additional Remarks:
   - The different positions of the minima of the signal intensity in the experimental and simulation results is never discussed.
   - I think the authors should put more emphasis on distinguishing coherent from incoherent effects that lead to signal decay. Since the experimental data clearly shows an oscillating behavior and no stochastic processes (e.g. molecular motion) are taken into account in the numerical simulations that are used to explain the observed phenomenon, the underlying mechanism has to be coherent. However, the abstract of the paper suggests "maxima in relaxation rates" (p. 1, l. 15) and "enhanced transverse relaxation" (p.1, l.10), which makes it sound like the observed signal decay is due to incoherent line broadening.
   - In the experimental data, it is not clear what the "signal intensity" is that is plotted. Is this the integrated intensity or the maximum peak intensity?
   - In the text and the figure legends the spinlock amplitude is referred to as nu_SL whereas in the axis of the figure it is denoted by nu_RF. The nomenclature should be consistent to avoid confusion.
   - The discussion of other reports of the effects of the MAS modulation of the rf field due to the spatial inhomogeneity is rather minimalistic. The authors may want to include the seminal works by Tekely and Goldman for example.
       o Goldman, M. and Tekely, P.: Effect of radial RF field on MAS spec tra, CR Acad. Sci. II C, 4, 795–800, 2001.
       o Tekely, P. and Goldman, M.: Radial-field sidebands in MAS, J. Magn. Reson., 148, 135–141, 2001.

References:

Tošner, Z., Purea, A., Struppe, J. O., Wegner, S., Engelke, F., Glaser, S. J., and Reif, B.: Radiofrequency fields in MAS solid state NMR probes, J. Magn. Reson., 284, 20–32, https://doi.org/10.1016/j.jmr.2017.09.002, 2017.

Gupta, R., Hou, G., Polenova, T., and Vega, A. J.: RF Inhomogeneity and how it Control CPMAS, Solid State Nucl. Magn. Reson., 72, 17–26, https://doi.org/10.1016/j.ssnmr.2015.09.005, 2015.

Hoult, D. I. and Richards, R. E.: The signal-to-noise ratio of the nu clear magnetic resonance experiment, J. Magn. Reson., 24, 71 85, https://doi.org/10.1016/0022-2364(76)90233-X, 1976.

---

## Author Comment (AC2)

We are grateful to the reviewers for their time and effort in reviewing our manuscript. Since several similar comments were raised by all the reviewers, below is a short response to the main comments of all reviewers:

- Spin-lock profile of $^{1}$H in 99.96% $D_2O$ were acquired and added to the main text.
- The discussion regarding the simulations was extended. In particular, numerical simulations were compared with the first-order Hamiltonian solution.
- We added more simulations and discussion to emphasize that the appearance of rotary-resonance conditions in isotropic samples is not due to the amplitude or width of the distribution (either external magnetic field or radio-frequency field irradiation). While the amplitude and width of the distribution influence the width of the rotary-resonance conditions, the modulation frequency, and the minimum intensity of the spin-lock signal at these conditions, the rotary-resonance conditions arise due to the periodic dependence.

Below is a point-by-point response (in blue) to reviewer comments (in black).

**Anonymous Referee #1**

1) Numerical simulations

The description of the numerical simulations provided in the SI suggests that the authors performed simulations for a single-spin-1/2 system by numerical integration of the Liouville-von Neumann equation. Based on Eq. (S2), both the initial density operator as well as the detection operator correspond to Ix (x magnetization). For simulations taking the spatial rf-field inhomogeneity into account, the Hamiltonian of the system corresponds to a spinlock along the x-axis (Eq. (S3C)) and an orthogonal component along the y-axis that is modulated by the MAS frequency (Eq. (S3B)). This will generate an effective field whose orientation and magnitude will be modulated by the MAS frequency and about which the magnetization will nutate. Due to spatial inhomogeneity of the rf field, the orientation and magnitude of this effective field will vary across the sample space. The authors model the spatial rf-field distribution using distribution functions f_B1 (that models the amplitude distribution along the rotor axis) and f_nu1 (that models the magnitude of the modulated component along the y-axis) and compute the total signal as the weighted average of these distributions (Eq. (S2)). Unfortunately, the functional forms of these distribution functions are not provided (which they should be) and can only be approximated based on Fig. S1. Since no other spin-spin or spin-field interactions are mentioned, it must be assumed that no other interactions are taken into account, which would be in line with an isotropic sample of an isolated nucleus. Moreover, the authors explicitly say that no relaxation is taken into account in the simulations and the effect therefore has to be coherent in nature. However, computing the signal based on this information (which is computationally inexpensive for a single-spin system) does not lead to the same results presented by the authors in Fig. 3 in the main text. I would therefore ask the authors to provide additional information on how the simulations were performed.

We have added the Table S1 to the supplementary information (SI), which summarizes the used weighting factors:

| | The Amplitude (in kHz) | The weighting factor |
|---|---|---|
| $B_0$ modulation $k$=[1:51] | $0.2(k-1)/V$ | $2(k-1)/V^2$ |

| | | |
|---|---|---|
| $B_1$ modulation, time-independent $l=[1:29]$ | $v_{SL}\left(1-(l-1)\dfrac{0.05}{M}\right)$ | $e^{-\left(1.89\frac{(l-1)}{M}\right)^2}\bigg/\left(\displaystyle\sum_{l=1}^{M}e^{-\left(1.89\frac{(l-1)}{M}\right)^2}\right)$ |
| $B_1$ modulation, time-dependent $k=[1:29]$ | $0.1v_{SL}\left(1-e^{-\left(1.26\frac{(k-1)}{W}\right)^4}\right)$ | $e^{-\left(1.26\frac{(k-1)}{W}\right)^4}\bigg/\left(\displaystyle\sum_{k=1}^{W}e^{-\left(1.26\frac{(k-1)}{W}\right)^4}\right)$ |

**Table S1** The summary of the amplitudes and the weighting factors for $B_0$ and $B_1$ modulations (the simulations are shown in Figures 4 and 5 in the main text). In simulations, M=29, W=29 and V=51.

We have also extended our discussion regarding the simulations and added the first-order Hamiltonian solution, comparing it with the numerical simulations. The following paragraph has been included in SI:

*"To understand the origin of the pseudo RRD effect, we start with the simplest case, investigating the behavior of a single on-resonance spin during the rf-field spin-lock ($H_{SL}$). The single spin inside the coil may be affected by an additional time-periodic term ($H_t$), orthogonal to the applied rf-field spin-lock. For simplicity, we also do not include any relaxation effects.*

*This additional term can depend on the external magnetic field ($B_0$ modulation), or the strength of the applied RF-field spin-lock ($B_1$ modulation). In solid samples, it can arise due to anisotropic interactions. The first and second modulations are related to inhomogeneities in the external magnetic field and RF field, respectively. The third could arise in liquid-like samples if there is some degree of alignment and therefore residual anisotropic interactions present. Knowing the exact values of these modulations (and shapes in the case of the distribution) is important when their influence is investigated quantitatively for a specific coil. Our goal here is the qualitative determination of the source of the pseudo RRD effect in the experiments.*

*In all three cases, the total Hamiltonian for this spin can be described as follows:*

$$H_{total} = H_{SL} + H_t = \omega_{SL}I_x + 2\pi\sum_n a_n\cos(n\omega_R t + \phi_n)I_z\widehat{Op}, \qquad \text{Eqn. (S1)}$$

*where $\omega_{SL} = 2\pi v_{SL}$. Here, $\widehat{Op} = 1$ for a single spin or $\widehat{Op} = 2S_z$ for a two-spin system. While for anisotropic interactions, n is 1 or 2,[1,2] for $B_0$ and $B_1$ modulations, n may take any integer value.[3] This is because these modulations are not purely sinusoidal, there are contributions from overtone frequencies. In the experimental SL profiles (Figures 2 and 3 in the main text), two rotary-resonance conditions are clearly observed. Therefore, in the following discussion, $n = 1, 2$ will be considered for all three cases.*

*The simulated SL-signal is defined as follows:*

$$S_{SL}(t_{SL}) = Tr\left\{I_x\widehat{T}e^{-i\int_0^{t_{SL}}dt\,H_{total}}I_x\widehat{T}e^{i\int_0^{t_{SL}}dt\,H_{total}}\right\}, \qquad \text{Eqn. (S2)}$$

*where $\widehat{T}$ is a Dyson operator. To simplify Eqn. (S1), the total Hamiltonian is transformed into the tilted rf-field frame:[1]*

$$H_{tot}^{rf} = U_{SL}^{-1}H_{tot}(t)U_{SL} - H_{SL}, \qquad \text{Eqn. (S3)}$$

*where $U_{SL} = e^{-i\omega_{SL}t I_x}$ is a propagator. The modified Eqn. (S2) in the titled frame is written as follows:*

$$S_{SL}(t_{SL}) = Tr\left\{I_x\widehat{T}e^{-i\int_0^{t_{SL}}dt\,H_{tot}^{rf}}I_x\widehat{T}e^{i\int_0^{t_{SL}}dt\,H_{tot}^{rf}}\right\}, \qquad \text{Eqn. (S4)}$$

*since the initial and the measured operators ($I_x$) commute with $U_{SL}$.*

*The modified Eqn. (S1) is:*

$$H_{tot}^{rf} = 2\pi \sum_{n=1}^{2} a_n \cos(n\omega_R t + \phi_n)\left(I_z \cos(\omega_{SL}t) + I_y \sin(\omega_{SL}t)\right)\widehat{Op}, \qquad \text{Eqn. (S5)}$$

*where $\widehat{Op}$ remains unchanged as it commutes with $U_{SL}$. The Eqn. (S5) can be rewritten in the following way:*

$$H_{tot}^{rf} = \qquad\qquad\qquad\qquad \text{Eqn. (S6)}$$

$$\pi \sum_{n=1}^{2} a_n \left[\left(\cos\left((n\omega_R + \omega_{SL})t + \phi_n\right) + \cos\left((n\omega_R - \omega_{SL})t + \phi_n\right)\right)I_z + \right.$$
$$\left.\left(\sin\left((n\omega_R + \omega_{SL})t + \phi_n\right) - \sin\left((n\omega_R - \omega_{SL})t + \phi_n\right)\right)I_y\right]\widehat{Op}.$$

*We see in Eqn. (S6) both $I_z\widehat{Op}$ and $I_y\widehat{Op}$ operators, which do not commute with the initial and final operators and are cosine or sine modulated. For small an, these terms can be approximated as zero, except for specific values of the spin lock frequency.*

*Under specific cases, when $k\omega_R - \omega_{SL} = 0$ (k=1 or 2), Eqn. (S6) can be rewritten as:*

$$H_{tot}^{rf} = \pi a_k\left[\cos(\phi_n)I_z - \sin(\phi_n)I_y\right]\widehat{Op} + H_{else}^{rf}, \qquad \text{Eqn. (S7)}$$

*while $H_{else}^{rf}$ is:*

$$H_{else}^{rf} = \pi \sum_{n=1}^{2} a_n \left[\cos\left((n+k)\omega_R t + \phi_n\right)I_z + \sin\left((n+k)\omega_R t + \phi_n\right)I_y\right]\widehat{Op} \qquad \text{Eqn. (S8)}$$
$$+ a_j\left[\cos((-1)^{k+1}\omega_R t + \phi_n)I_z - \sin((-1)^{k+1}\omega_R t + \phi_n)I_y\right]\widehat{Op},$$

*where for the k=1 condition, j=2; and for the k=2 condition j=1. Eqn. (S7) can be further simplified using average Hamiltonian theory,[4] considering only the first-order term:*

$$T_R H_{tot}^{rf\,(0)} = \pi T_R a_k\left[\cos(\phi_n)I_z - \sin(\phi_n)I_y\right]\widehat{Op} = e^{-i\phi_n I_x}\frac{\pi a_k}{\nu_R}I_z\widehat{Op}e^{i\phi_n I_x}, \qquad \text{Eqn. (S9)}$$

*where the average, $H_{tot}^{rf\,(0)}$, is taken over one rotor period ($T_R = \frac{1}{\nu_R} = \frac{2\pi}{\omega_R}$). Regardless of the explicit form of the $\widehat{Op}$ operator, the measured spin-lock signal, according to Eqns. (S4) and (S9) is as follows:*

$$S_{SL}(t_{SL} = N_{SL}T_R) \approx \cos\left(\pi\frac{a_k}{\nu_R}N_{SL}\right), \qquad \text{Eqn. (S10)}$$

*since $e^{-i\phi_n I_x}$ commutes with the initial and final operators. For dipolar interactions, Eqn. (S10) should be modified to account for all orientations:*

$$S_{SL}(t_{SL} = N_{SL}T_R) \approx \int d\Omega \cos\left(\pi\frac{a_k}{\nu_R}N_{SL}\right), \qquad \text{Eqn. (S11)}$$

*The integration over orientation ($\Omega$) indicates the powder averaging with Euler angles, $(\alpha, \beta, \gamma)$.[1]*

    *Figure S1 compares numerical (solid lines) and FOH curves (stars) under the rotary-resonance conditions $\nu_{SL} = \nu_R$ (Figures S1A-C) and $\nu_{SL} = 2\nu_R$ (Figures S1D-F) and different $a_k$ values, related either to $B_0$ modulation (Figures S1A and D), dipolar interaction (two-spin system, Figure S1B and E) or $B_1$ modulation (Figures S1C and F). These figures show full agreement between numerical and FOH curves. In all three cases, the changes in $a_k$ values affect the modulation frequency of the spin-lock signal.*

[Figure]

***Figure S1*** *Numerical spin-lock (solid) and FOH (stars, Eqns. (S10) and (S11)) signals were simulated with different values of $B_0$ modulation (a single spin, A and D), dipolar coupling values (two spin-system, B and E) and $B_1$ modulation (a single spin, C and F) at rotary-resonance conditions, where $\nu_{SL} = \nu_R$ (A-C) and $\nu_{SL} = 2\nu_R$ (D-F). In (A) and (D), $a_2 = 0.5a_1$ with $a_1$: 0 – black; 200 Hz – red; 400 Hz – blue and 737 Hz – cyan. In (B) and (D), $a_1 = \frac{\nu_D}{\sqrt{2}}\sin(2\beta)$ and $a_2 = -\frac{\nu_D}{2}\sin^2(\beta)$ with dipolar coupling values $\nu_D$ of: 0 – black; 200 Hz – red; 400 Hz – blue and 737 Hz – cyan. In (C) and (F), $a_2 = 0.25a_1$ with $a_1$: 0 – black; 0.5% – red; 1% – blue and 1.37% – cyan. All simulations performed at 10 kHz MAS.*

Dependence of the numerical (solid lines) and FOH curves (stars) on MAS rate is shown in Figure S2 under the conditions $a_1 = 2a_2 = 0.2$ kHz of $B_0$ modulation (Figures S2 A and D), $\nu_D = 0.2$ kHz of dipolar interaction (Figures S2 B and E) and $a_1 = 4a_2 = 1$ % of nominal $B_1$ (Figures S2 C and F), for both rotary-resonance conditions ($\nu_{SL} = \nu_R$ and $\nu_{SL} = 2\nu_R$). Only for $B_1$ modulation (Figures S2 C and F) does the change of MAS rate affect the modulation frequency. This is a simple consequence of $B_1$ modulation amplitude scaling up with $B_1$.

[Figure]

*__Figure S2__ Numerical spin-lock (solid) and FOH (stars, Eqns. (S10) and (S11)) signals were simulated for $B_0$ modulation (a single spin, A and D), dipolar coupling values (two spin-system, B and E) and $B_1$ modulation (a single spin, C and F) at rotary-resonance conditions, where $\nu_{SL} = \nu_R$ (A-C) and $\nu_{SL} = 2\nu_R$ (D-F) under different $\nu_R$: 10 kHz – black; 20 kHz – red; and 35 kHz – blue.*

*While Figure S2 shows spin-lock signals only at rotary-resonance conditions, Figure S3 presents numerical SL profiles for spin-lock strengths between 1 and 44 kHz under three different MAS rates: 10 kHz (A-C), 20 kHz (D-F) and 35 kHz (G-I) for $B_0$ modulation (A, D and G), dipolar interaction (B, E and H) and $B_0$ modulation (C, E and I). The same conclusions as in Figure S2 are observed: for $B_0$ modulation (A, D and G) and dipolar interaction (B, E and H), the changes in MAS do not affect the modulation frequency, while for $B_1$ modulation (C, F and I), this is not the case (marked in gray in Figure S3). Additionally, the profiles show that the rotary-resonance conditions are narrow: a deviation of only 100 Hz from these conditions almost completely removes the influence of the time-dependent term on the spin-lock signal in all figures.*

[Figure]

**Figure S3** *Numerical SL profiles showing the influence of time dependence introduced via $B_0$ modulation ($a_1 = 0.5a_2 = 100$ Hz, A, D and G), dipolar interaction ($\nu_D = 200$ Hz, B, E and H) and $B_1$ modulation ($a_1 = 0.25a_2 = 0.01\nu_{SL}$, C, F and I). The simulated signal is shown as a function of the rf-field strength ($\nu_{SL}$, axis y) and mixing time ($t_{SL}$, axis x) of the SL under three different MAS rates: 10 kHz (A-C), 20 kHz (D-F) and 35 kHz (G-I). The values in gray represent the coordinates of the first minimum in the profiles. No phenomenological relaxation was included in the simulations.*

In addition to time dependence induced by $B_1$ inhomogeneity, there is also time independent inhomogeneity,[5–10] which is most clearly seen along the axis of the rotor. Including this in the simulation broadens the rotary-resonance conditions. With addition of a time-independent term, parallel to the applied spin lock Hamiltonian becomes:

$$H_{SL} = 2\pi\left(\nu_{SL}I_x - \Delta_{B_1}\nu_{SL}I_x\right) = 2\pi\nu_{SL}\left(1 - \Delta_{B_1}\right)I_x, \qquad \textit{Eqn. (S12)}$$

*where $\Delta_{B_1}$ represent the inhomogeneity factor for a given position in the sample. Experimentally, the reason of the appearance of this term is due to the fact that a solenoidal coil produces a higher rf-field at the center, as compared with the edges. Rather than explicitly averaging over the sample volume, we consider an approximate linear distribution of spin-lock signal. The average signal is then the sum of M signals with $\nu_{SL,l} = \nu_{SL}\left(1 - (l-1)\frac{\Delta_{B_1}}{M}\right) (l = 1, \dots, M)$ and normalized with a Gaussian weighting factor,[5,6,8,11] $g_{B_1,l}$:*

$$S_{SL}(t_{SL}) = \sum_{l=1}^{M} g_{B_1,l}\, Tr\left\{I_x \hat{T} e^{-i\int_0^{t_{SL}} dt H_{total,l}} I_x \hat{T} e^{i\int_0^{t_{SL}} dt H_{total,l}}\right\}, \qquad \text{Eqn. (S13)}$$

*where $\hat{T}$ is a Dyson operator and $g_{B_1,l} = e^{-\left(1.89\frac{(l-1)}{M}\right)^2} / \left(\sum_{l=1}^{M} e^{-\left(1.89\frac{(l-1)}{M}\right)^2}\right)$.*

*The total Hamiltonian, $H_{total,k}$, is defined as follows:*

$$H_{total,l} = \omega_{SL}\left(1 - (l-1)\frac{\Delta_{B_1}}{M}\right) I_x + 2\pi \sum_{n=1}^{2} a_n \cos(n\omega_R t + \phi_n) I_z \widehat{Op}. \qquad \text{Eqn. (S14)}$$

*In the following simulations, $\Delta_{B_1}$=0.05 (5% with respect to applied $\nu_{SL}$). While the value of $\Delta_{B_1}$ as well as the weighting factor can be different from coil to coil, this term is not the source of the appearance rotary-resonance conditions. However, this term broadens the rotary-resonance conditions and alters the positions and values of the first minimum signal intensities, as shown in Figure S4.*

*As in the previous simulations, for $B_0$ modulation (A, D and G) and dipolar interaction (B, E and H), changes in MAS do not affect the modulation frequency, while for $B_1$ modulation (C, F and I), this is not the case. Additionally, for both $B_0$ modulation and dipolar interaction, the intensities at the first minima show a dependence on MAS rate, while for $B_1$ modulation, they do not (marked in gray in Figure S4). This differing dependence on MAS rate for $B_1$ modulation versus $B_0$ modulation and dipolar interaction could indicate which effect plays a major role in rotary-resonance condition experiments for isotropic samples. Since the rotary-resonance effect is observed for $^1H$ spins in 99.96% $D_2O$ (Figure 3A-B in the revised main text), the dipolar interaction was not included in simulations in the main text.*

[Figure]

***Figure S4*** *Numerical SL profiles with an additional time-independent term ($\Delta_{B_1}$ = 0.05, Eqns. S12-S14) showing the influence of time dependence introduced via $B_0$ modulation ($a_1 = 0.5a_2 = 100$ Hz, A, D and G), dipolar interaction ($\nu_D = 200$ Hz, B, E and H) and $B_1$ modulation ($a_1 = 0.25a_2 = 0.01\nu_{SL}$, C, F and I). The simulated signal is shown as a function of the rf-field strength ($\nu_{SL}$, axis y) and mixing time ($t_{SL}$, axis x) of the SL under three different MAS rates: 10 kHz (A-C), 20 kHz (D-F) and 35 kHz (G-I). The values in gray represent the coordinates of the first minimum in the profiles. No phenomenological relaxation was included in the simulations."*

2) Model for the rf inhomogeneity

In my opinion, the model used for the spatial distribution of the rf-field in a solenoid coil is not appropriate due to the following issues: - The authors assume an amplitude deviation from the nominal spinlock amplitude of only 0-5% (Eq. (S3C)). However, both simulated (e.g. Tosner et. al. 2017) and measured (e.g. Gupta et. al. 2015) rf-field distributions of common MAS NMR probes suggest that the field drops to approximately 50% at the edges of the sample space. Since there is no mention of spatial sample restriction to the central part of the rotor, the chosen amplitude distribution is not appropriate. -

For a maximum deviation of the spinlock amplitude of 5% the signal weighting function given in Fig. S1C is not appropriate. According to the reciprocity theorem (e.g. Hoult 1976), the induced signal is directly related to the B1 field of the receiving coil at a given position in the sample space. A deviation from the nominal field by 5% should therefore not lead to a weighing of the signal with a factor of close to zero as is shown in Fig. S1C. - The orthogonal component (Eq. (S3B)) occurs due to the MAS modulation of the phase of the rf irradiation which is known to be strongest at the edges of the sample space (e.g. Tosner 2017). Since the spinlock amplitude is also significantly lower than the nominal spinlock amplitude in these parts of the sample, the rotary resonance condition is no longer fulfilled in these parts of the sample and the coil sensitivity is significantly lower. However, the model implemented by the authors uses the same magnitude of the modulation for all values of f_B1. Since simulated realistic rf-field distributions can be found in the literature (Tosner et. al. 2017), I would suggest that the authors perform their simulations using such a distribution instead of this crude model that

The following figure depicts the Fourier Transform of the nutation curve (black), which was acquired using a series of 1D $^1$H experiments in 99.96% $D_2O$ as a function of the single pulse length, ranging from 0.5 μs to 160 μs in 0.5 μs steps. The Gaussian function (red dashed line) with a standard deviation of 0.027μ (μ = 79.5 kHz) shows that this distribution can approximately be considered Gaussian (as reported in previous works),[4–6] and that the 5% maximum deviation used in the simulations approximates the actual distribution. The exact distribution would only be necessary for quantitative comparison with experiments.

[Figure]

Fast Fourier Transform (FFT) of 1D $^1$H signal as a function of the length of single proton pulse (black) and comparison with a Gaussian function (red dashed). The sample was 99.96% $D_2O$. For FFT, MATLAB software was used. The experiments were performed at 10 kHz MAS.

3) Isotropic sample

The experimental results presented in the paper stem from measurements of a polybutadiene rubber sample as well as a polyethylene glycol sample. Although these sample exhibit "liquidlike" spectra, they

both correspond to large polymer systems, where partial alignment might lead to residual anisotropic dipolar interactions that will be recoupled for irradiation at a rotary resonance condition. The comparison of experimental spectra of the rubber sample with and without MAS (Fig. S4A) shows that the static spectrum is significantly broader than the one under MAS. This line broadening can either be attributed to residual anisotropic interactions or susceptibility effects and raises the question if either of the two are re-introduced by irradiating at the rotary resonance condition which would lead to signal decay. The authors attribute the observed signal decay solely to the modulation of the rf-field due to the rf inhomogeneity, based on numerical simulations of a single-spin system. This raises the question why the authors didn't choose to demonstrate the effect using a truly isotropic system (such as, for example, the residual H2O line in D2O) that would correspond more closely to what is simulated.

The suggested experiments were performed and included into the revised version of the article as Figure 3. We also observed rotary-resonance conditions for $^1$H in 99.96% $D_2O$.

[Figure]

$^1$H in 99.96% $D_2O$ experiments. (A) 1D spectrum recorded at 10 kHz MAS. (B) $^1$H signal is shown as a functions of the rf-field strength ($\nu_{SL}$, y-axis) and mixing time ($t_{SL}$, x-axis) of the SL pulse. The sample contained $Cu^{2+}$ ethylenediaminetetraacetic acid disodium salt (5 mM) to accelerate the acquisition.

4) Additional Remarks:

- The different positions of the minima of the signal intensity in the experimental and simulation results is never discussed.

The positions of the signal intensity minima in the experimental and simulated SL profiles are similar. In the experimental profiles, we observe minima at approximately 2.4 ms, 2.4 ms, 1 ms, and 0.4 ms for 10 kHz, 10 kHz half-windowed, 25 kHz, and 35 kHz MAS, respectively. In the simulations, these values are 1.8 ms, 1.8 ms, 0.9 ms, and 0.5 ms. In the experiments, the minima positions are broader than in the simulations. For example, for 10 kHz MAS (Figure 2A), the minima can be identified between 1.8 ms and 3 ms, where similar minima intensities are observed. The intensity values at these minima points depend on the amplitude of the $B_1$-modulated term. Since our goal is a qualitative comparison of the experimental and simulated SL profiles, we do not expect a full agreement between them.

- I think the authors should put more emphasis on distinguishing coherent from incoherent effects that lead to signal decay. Since the experimental data clearly shows an oscillating behavior and no stochastic

processes (e.g. molecular motion) are taken into account in the numerical simulations that are used to explain the observed phenomenon, the underlying mechanism has to be coherent. However, the abstract of the paper suggests "maxima in relaxation rates" (p. 1, l. 15) and "enhanced transverse relaxation" (p.1, l.10), which makes it sound like the observed signal decay is due to incoherent line broadening.

We corrected the mentioned sentence:

"*A drastic reduction in spin-lock signal intensities is observed when the spin-lock frequency matches one or two times the MAS rate. Under these conditions, the spin-lock signal decays at a much higher rate, and oscillations of the signal are observed, consistent with a coherent origin of the effect.*"

A comparison with the first-order Hamiltonian solution (please refer to the response to point #1) emphasizes that we are considering only coherent effects.

- In the experimental data, it is not clear what the "signal intensity" is that is plotted. Is this the integrated intensity or the maximum peak intensity?

The peak maximum was used to plot all SL profiles, except for one in Figure S4 of the supplementary information (first version), where both the integrated intensities and peak intensities were measured from the same experimental data. We have added clarification to the figure captions.

- In the text and the figure legends the spinlock amplitude is referred to as nu_SL whereas in the axis of the figure it is denoted by nu_RF. The nomenclature should be consistent to avoid confusion.

Corrected.

- The discussion of other reports of the effects of the MAS modulation of the rf field due to the spatial inhomogeneity is rather minimalistic. The authors may want to include the seminal works by Tekely and Goldman for example. o Goldman, M. and Tekely, P.: Effect of radial RF field on MAS spec tra, CR Acad. Sci. II C, 4, 795–800, 2001. o Tekely, P. and Goldman, M.: Radial-field sidebands in MAS, J. Magn. Reson., 148, 135–141, 2001.

We have added the citation of the mentioned works with the following discussion. In addition we cite Aebischer et al. MR 2021, in which radial spatial inhomogeneity is investigated.

"*Some other works regarding the effects of radial RF field on MAS spectra have been reported previously,[3,9,12,13] where the appearance of sidebands due to time-dependent rf-field inhomogeneity was theoretically explored and experimentally demonstrated. Particularly, Aebischer et al.[3] investigated the influence of time-dependent modulations of the rf-field amplitude and phase on the performance of selected solid-state NMR experiments. The influence of both amplitude and phase modulations was observed in samples with non-vanishing anisotropic interactions. These modulations did not significantly affect most recoupling sequences, with the exception of double quantum C-symmetry sequences.[14] Consistent with the matching conditions identified in this study, Aebischer et al.[3] revealed significant amplitudes at $\nu_R$ and $2\nu_R$ in nutation spectra.*

**Referee #2: Zdeněk Tošner**

This contribution reports about a surprising observation that T1rho of liquid-like samples shows enhanced relaxation at rotary resonance conditions known from rotating solids. It is not expected for samples where dipolar interactions are completely averaged by fast motions. It is argued that this phenomenon arises when liquid-like samples are rotated in a spatially inhomogeneous radiofrequency field of solenoid coils. Consequently, similar effect could be expected for solids, changing thus interpretation of rotary resonance relaxation dispersion experiments.

While I understand urgency of this report, I would appreciate a more thorough study. Here are my concerns:

- Rubber and polymer samples are liquid-like, but not quite typical liquids. There still might be nonzero residual dipolar couplings. MAS imposes centrifugal forces and polymer particles may become "solidified" on the rotor walls – can it be excluded?

We also observed rotary-resonance conditions for $^1$H in 99.96% $D_2O$. See also the response to point #3 from Reviewer 1.

- If the explanation is solely time-variable rf field, it can be modelled in liquid state probes without sample spinning by applying shaped pulses. These experiments would then be very convincing experimental proof of the phenomenon.

We have simulated the effect as fluctuations in the z-component of the rf field generated by a solenoid tilted at the magic angle. (The y-component can also be significant.) The situation is quite different in a solution probe, where generation of a z-component would not be achievable with the coil. Yes, a y-component could be generated with a digitized shape pulse. However, this artificial setup would not add much to what is already evident from the simulations, in our view. It would not tell us whether $B_0$ or $B_1$ fields are more significant in the MAS rotor.

- Quality of the supporting material is not sufficient to reproduce the simulations. There are many typos and unclear nomenclature. For example, what is integral of "df dg f(x) g(B1)" (what is the "df" element?) The time-variable rf Hamiltonians are defined using a sum over n (n=1, 2) of cos(\omega_R t) – there is no "n" under the sum. A graph of these Hamiltonians would help. No reasoning is given for f(\nu_1) being biquadrate exponential distribution function, etc…

For simulation details, please refer to the response to point #1 from Reviewer 1. We thank the reviewer for pointing out the missing 'n' in Eqn. (S3). We added the sentence regarding the biquadrate exponential distribution function:

"*The bi-quadrate exponential distribution function does not represent the real distribution of the time-dependent $B_1$ modulation; it simply provides non-linear sampling, different from the time-independent term.*"

- I miss more detailed simulation study on the extent of rf modulations introducing enhanced relaxation. I mean some assessment what the amplitude of rf modulations must be, perhaps in relation to the nominal rf amplitude, to observe the phenomenon? RF

field modulations within an MAS rotor are of different amplitude, depending on the position along the coil axis as well as on the radial distance… In simulations, 10% amplitude modulations are assumed regardless of the position.

We agree that more a detailed simulation study is important, when this effect is quantitatively investigated for a specific coil. However, it is beyond the scope of this article, as our goal is to illustrate the effect. Our approach does not explicitly consider radial distance. However the inclusion of a distribution of amplitudes for the modulation does approximate the situation in the sample.

- Our detailed calculation of rf field in solenoids show MAS induced variations on the frequency of \omega_R. Where exactly is the origin of the second recoupling condition at two times \omega_R? It comes naturally from MAS modulations of the residual dipolar couplings (rank-2 tensors) but rf field is modulated at just a single frequency…

The amplitude and phase profiles in the previously reported work[9] are not purely sinusoidal. This makes sense when considering a location in the rotor that passes close to a coil turn over a small angular distance, and is otherwise far from that coil turn, and is particularly noticeable towards the edge of the coil. Since the wave is not purely sinusoidal, there are contributions from overtone frequencies (see e.g. sawtooth or triangular waves, in which many overtones are present). We kept only n=1 and n=2, since these are clear experimentally. However, rotary-resonance conditions with n>2 may also be observed. n=3 appears to show up very weakly in the new $D_2O$ data, in which we sampled more carefully around this condition.

**Anonymous Referee #3**

- Would one expect such behavior for water or generally for molecules tumbling in the picosecond time scale, true "liquid-like," for example, glycine in water? Is there any specific reason for restricting the study to PEG and rubber? The molecular weight of the PEG used in the study is unclear. One would also expect high molecular weight PEGs to show anisotropic molecular tumbling or sediment on the rotor wall. Perhaps data on molecules showing averaging of dipolar interactions would be comprehensive proof.

We also observed pseudo rotary-resonance conditions for [1]H in 99.96% $D_2O$. See also the response to point #3 from Reviewer 1.

- Perhaps it might be helpful to measure the rotation correlation timescale of the PEG sample instead of making a hand-waving argument.

Since we see the effect in the residual water in $D_2O$, this should not be necessary (the effect is not dependent on the change of rotational correlation times between PEG and water).

- The decay of magnetization appears (not sure) to indicate minima/oscillations. Is this a feature of incoherent effects?

This is a characteristic of the coherence effects. We corrected the sentences that previously mistakenly implied incoherent effects were being considered.

- Is there any special reason for using different decoupling schemes at different MAS frequencies, especially if the claim is that the samples are "solution-like"? It's probably not a relevant point, but it's just curiosity.

    The same decoupling scheme was used at different MAS frequencies. However, different decoupling schemes were used for different probes: WALTZ16[15] for the 1.3 mm and SW$_f$-TPPM[16] for the 4 mm. There was no particular reason for this; it is just the result of setting up the sequences for the different probes, and selecting a decoupling sequence.

- Comparing Figure 2 and Figure 4, despite using smaller rf inhomogeneity in simulations compared to typically reported on MAS probes, the magnetization decays substantially faster in simulations. What is the experimental rf inhomogeneity of the rf field?

    Please refer to the response to point #4 from Reviewer 1. The key feature of the inhomogeneity has to do with the oscillatory part, which is not easy to characterize experimentally.

- The simulation details are sketchy, especially the terms in Eq S2.

    Please refer to the response to point #1 from Reviewer 1.

- By naively looking at Equations S3A and S3B, it's not so apparent why the simulations differ except for the weighting factors, which have a different slope and magnitude. In Eq S3, "n is missing" in the cosine term. Is there a justification for summing over n=1,2, or is it motivated by experimental observations?

    We thank the reviewer for pointing out the missing 'n' in Eqn. (S3). The summing over n=1, 2 was motivated by experimental observation, despite the possibility for higher values of n to contribute.

- Is it sufficient to consider inhomogeneity distribution along one axis in both scenarios? Why should that be the case?

    We have chosen not to explicitly simulate over the spatial axes of the sample, but rather provide a qualitative simulation. Quantitative agreement would require a more detailed simulation, and also taking precautions that there is little to no air bubble present after sample loading, otherwise the very center of the rotor will not be filled with sample. (In case the reviewer refers to spin axes, in revised version we show simulations, in which we consider $B_0$ and $B_1$ modulations along the same spin axis ($I_z$).)

Minor points

- Page 3, line 9 chose

We are not sure what the reviewer means. With the addition of the water sample, this text now reads:

*"The same behavior is observed for a polyethylene glycol solution at 10 kHz MAS and for residual protons in liquid deuterium oxide.*

*We chose these samples since the polybutadiene rubber displays liquid-like spectra but does not undergo translational diffusion due to the elastomeric properties of a cross-linked polymer. On the other hand, since the polybutadiene is an elastomer and therefore may not undergo perfect isotropic averaging, we also recorded data for a polyethylene glycol solution and liquid water."*

- Different figures have different labels for the rf axis.

    Corrected.

(1) Mehring, M. *Principles of High Resolution NMR in Solids*, 2nd ed.; Springer-Verlag: Berlin Heidelberg, 1983. https://doi.org/10.1007/978-3-642-68756-3.
(2) Olejniczak, E. T.; Vega, S.; Griffin, R. G. Multiple Pulse NMR in Rotating Solids. *J. Chem. Phys.* **1984**, *81* (11), 4804–4817. https://doi.org/10.1063/1.447506.
(3) Aebischer, K.; Tošner, Z.; Ernst, M. Effects of Radial Radio-Frequency Field Inhomogeneity on MAS Solid-State NMR Experiments. *Magn. Reson.* **2021**, *2* (1), 523–543. https://doi.org/10.5194/mr-2-523-2021.
(4) Haeberlen, U.; Waugh, J. S. Coherent Averaging Effects in Magnetic Resonance. *Phys. Rev.* **1968**, *175* (2), 453–467. https://doi.org/10.1103/PhysRev.175.453.
(5) Engelke, F. Electromagnetic Wave Compression and Radio Frequency Homogeneity in NMR Solenoidal Coils: Computational Approach. *Concepts Magn. Reson.* **2002**, *15* (2), 129–155. https://doi.org/10.1002/cmr.10029.
(6) Gupta, R.; Hou, G.; Polenova, T.; Vega, A. J. RF Inhomogeneity and How It Control CPMAS. *Solid State Nucl. Magn. Reson.* **2015**, *72*, 17–26. https://doi.org/10.1016/j.ssnmr.2015.09.005.
(7) Hoult, D. I. Solvent Peak Saturation with Single Phase and Quadrature Fourier Transformation. *J. Magn. Reson. 1969* **1976**, *21* (2), 337–347. https://doi.org/10.1016/0022-2364(76)90081-0.
(8) Paulson, E. K.; Martin, R. W.; Zilm, K. W. Cross Polarization, Radio Frequency Field Homogeneity, and Circuit Balancing in High Field Solid State NMR Probes. *J. Magn. Reson.* **2004**, *171* (2), 314–323. https://doi.org/10.1016/j.jmr.2004.09.009.
(9) Tošner, Z.; Purea, A.; Struppe, J. O.; Wegner, S.; Engelke, F.; Glaser, S. J.; Reif, B. Radiofrequency Fields in MAS Solid State NMR Probes. *J. Magn. Reson.* **2017**, *284*, 20–32. https://doi.org/10.1016/j.jmr.2017.09.002.
(10)   Tošner, Z.; Sarkar, R.; Becker-Baldus, J.; Glaubitz, C.; Wegner, S.; Engelke, F.; Glaser, S. J.; Reif, B. Overcoming Volume Selectivity of Dipolar Recoupling in Biological Solid-State NMR Spectroscopy. *Angew. Chem. Int. Ed.* **2018**, *57* (44), 14514–14518. https://doi.org/10.1002/anie.201805002.
(11)   Xue, K.; Nimerovsky, E.; Tekwani Movellan, K. A.; Becker, S.; Andreas, L. B. Backbone Torsion Angle Determination Using Proton Detected Magic-Angle Spinning Nuclear Magnetic Resonance. *J. Phys. Chem. Lett.* **2022**, *13* (1), 18–24. https://doi.org/10.1021/acs.jpclett.1c03267.

(12)    Tekely, P.; Goldman, M. Radial-Field Sidebands in MAS. *J. Magn. Reson.* **2001**, *148* (1), 135–141. https://doi.org/10.1006/jmre.2000.2215.

(13)    Goldman, M.; Tekely, P. Effect of Radial RF Field on MAS Spectra. *Comptes Rendus Académie Sci. - Ser. IIC - Chem.* **2001**, *4* (11), 795–800. https://doi.org/10.1016/S1387-1609(01)01310-X.

(14)    Lee, Y. K.; Kurur, N. D.; Helmle, M.; Johannessen, O. G.; Nielsen, N. C.; Levitt, M. H. Efficient Dipolar Recoupling in the NMR of Rotating Solids. A Sevenfold Symmetric Radiofrequency Pulse Sequence. *Chem. Phys. Lett.* **1995**, *242* (3), 304–309. https://doi.org/10.1016/0009-2614(95)00741-L.

(15)    Shaka, A. J.; Keeler, J.; Frenkiel, T.; Freeman, R. An Improved Sequence for Broadband Decoupling: WALTZ-16. *J. Magn. Reson. 1969* **1983**, *52* (2), 335–338. https://doi.org/10.1016/0022-2364(83)90207-X.

(16)    Thakur, R. S.; Kurur, N. D.; Madhu, P. K. Swept-Frequency Two-Pulse Phase Modulation for Heteronuclear Dipolar Decoupling in Solid-State NMR. *Chem. Phys. Lett.* **2006**, *426* (4), 459–463. https://doi.org/10.1016/j.cplett.2006.06.007.

---

## Author Response (AR1)

We are grateful to the reviewers for their time and effort in reviewing our manuscript. Since several similar comments were raised by all the reviewers, below is a short response to the main comments of all reviewers:

- Spin-lock profile of 1H in 99.96% D2O were acquired and added to the main text.
- The discussion regarding the simulations was extended. In particular, numerical simulations were compared with the first-order Hamiltonian solution.
- We added more simulations and discussion to emphasize that the appearance of rotaryresonance conditions in isotropic samples is not due to the amplitude or width of the distribution (either external magnetic field or radio-frequency field irradiation). While the amplitude and width of the distribution influence the width of the rotary-resonance conditions, the modulation frequency, and the minimum intensity of the spin-lock signal at these conditions, the rotary-resonance conditions arise due to the periodic dependence.

**Below is a point-by-point response (in blue) to reviewer comments (in black).**

**Anonymous Referee #1**

**1) Numerical simulations**

The description of the numerical simulations provided in the SI suggests that the authors performed simulations for a single-spin-1/2 system by numerical integration of the Liouville-von Neumann equation. Based on Eq. (S2), both the initial density operator as well as the detection operator correspond to Ix (x magnetization). For simulations taking the spatial rf-field inhomogeneity into account, the Hamiltonian of the system corresponds to a spinlock along the x-axis (Eq. (S3C)) and an orthogonal component along the y-axis that is modulated by the MAS frequency (Eq. (S3B)). This will generate an effective field whose orientation and magnitude will be modulated by the MAS frequency and about which the magnetization will nutate. Due to spatial inhomogeneity of the rf field, the orientation and magnitude of this effective field will vary across the sample space. The authors model the spatial rf-field distribution using distribution functions f\_B1 (that models the amplitude distribution along the rotor axis) and f\_nu1 (that models the magnitude of the modulated component along the yaxis) and compute the total signal as the weighted average of these distributions (Eq. (S2)). Unfortunately, the functional forms of these distribution functions are not provided (which they should be) and can only be approximated based on Fig. S1. Since no other spin-spin or spin-field interactions are mentioned, it must be assumed that no other interactions are taken into account, which would be in line with an isotropic sample of an isolated nucleus. Moreover, the authors explicitly say that no relaxation is taken into account in the simulations and the effect therefore has to be coherent in nature. However, computing the signal based on this information (which is computationally inexpensive for a single-spin system) does not lead to the same results presented by the authors in Fig. 3 in the main text. I would therefore ask the authors to provide additional information on how the simulations were performed.

We have added the Table S1 to the supplementary information (SI), which summarizes the weighting factors used. We also extended the discussion.

*"Time-dependent modulation may also be distributed. In that case, spin-lock signal will depend on the additional loop:*

$$S_{SL}(t_{SL}) = Eqn. (S15)$$

$$N_{f,g} \sum_{k=1}^{W_x} f_{x,k} \sum_{l=1}^{M} g_{B_{1,l}} Tr \left\{ I_x \hat{T} e^{-i \int_0^{t_{SL}} dt H_{total,(k,l)}} I_x \hat{T} e^{i \int_0^{t_{SL}} dt H_{total,(k,l)}} \right\},$$

where  $N_{f,g}$  is a normalization factor,  $x=B_1$  or  $B_0$ . The total Hamiltonian,  $H_{total,(k,l)}$ , is defined as follows:

$$H_{total,(k,l)} = Eqn. (S16)$$

$$\omega_{SL} (1 - G_{SL,l}) I_x + \{2\pi A_{x,k} \sum_{n=1}^2 a_n \cos(n\omega_R t + \phi_n)\} I_z \widehat{Op},$$
where  $x = B_1$  or  $B_0$ . Here,  $\widehat{Op} = 1$  for a single spin or  $\widehat{Op} = 2S_z$  for a two-spin system.

Table S1 summarizes the amplitudes and the weighting factors for all time-dependent ( $B_0$  and  $B_1$  modulations) and time-independent ( $B_1$ ) terms:

|                   | $a_n$ (in kHz)     | The Amplitude                                     | The weighting factor          |
|-------------------|--------------------|---------------------------------------------------|-------------------------------|
| $B_0$ modulation  | $a_1 = 2a_2 = 0.2$ | $A_{B_0,k} = (k-1)/25$                            | $f_{B_0,k} = (k-1)/25$        |
| k =[1:26]  |                    |                                                   |                               |
| $B_1$ modulation, |                    | 0.05                                              | $-(189(l-1))^2$               |
| time-             | -                  | $G_{SL,l} = (l-1) \frac{1}{28}$                   | $g_{B_1,l} = e^{(1.05 \ 28)}$ |
| independent       |                    |                                                   |                               |
| l=[1:29]   |                    |                                                   |                               |
| $B_1$ modulation, | $a_1 = 4a_2$       | $\left(-\left(126\frac{(k-1)}{k}\right)^4\right)$ | $-(126\frac{(k-1)}{2})^4$     |
| time-dependent    | $= 0.1 v_{SL}$     | $A_{B_{1},k} = \left(1 - e^{-(1.20 \ 28)}\right)$ | $f_{B_1,k} = e^{(1.20 28)}$   |
| k =[1:29]  |                    |                                                   |                               |

**Table S1** The summary of the amplitudes and the weighting factors for  $B_0$  and  $B_1$  modulations (the simulations are shown in Figures 4 and 5 in the main text). "

We have also extended our discussion regarding the simulations and added the first-order Hamiltonian solution, comparing it with the numerical simulations. The following paragraph has been included in SI:

"To understand the origin of the pseudo RRD effect, we start with the simplest case, investigating the behavior of a single on-resonance spin during the rf-field spin-lock ( $H_{SL}$ ). The single spin inside the coil may be affected by an additional time-periodic term ( $H_t$ ), orthogonal to the applied rf-field spin-lock. For simplicity, we also do not include any relaxation effects.

This additional term can depend on the external magnetic field ( $B_0$  modulation), or the strength of the applied RF-field spin-lock ( $B_1$  modulation). In solid samples, it can arise due to anisotropic interactions. The first and second modulations are related to inhomogeneities in the external magnetic field and RF field, respectively. The third could arise in liquid-like samples if there is some degree of alignment and therefore residual anisotropic interactions present. Knowing the exact values of these modulations (and shapes in the case of the distribution) is important when their influence is investigated quantitatively for a specific coil. Our goal here is the qualitative determination of the source of the pseudo RRD effect in the experiments.

In all three cases, the total Hamiltonian for this spin can be described as follows:

 $H_{total} = H_{SL} + H_t = \omega_{SL}I_x + 2\pi \sum_n a_n \cos(n\omega_R t + \phi_n)I_z \widehat{Op},$  Eqn. (S1) where  $\omega_{SL} = 2\pi v_{SL}$ . Here,  $\widehat{Op} = 1$  for a single spin or  $\widehat{Op} = 2S_z$  for a two-spin system. While for anisotropic interactions, n is 1 or 2,1,2 for  $B_0$  and  $B_1$  modulations, n may take any integer value.3 This is because these modulations are not purely sinusoidal, there are contributions from overtone frequencies. In the experimental SL profiles (Figures 2 and 3 in the main text), two rotary-resonance conditions are clearly observed. Therefore, in the following discussion, n = 1, 2 will be considered for all three cases.

The simulated SL-signal is defined as follows:

$$S_{SL}(t_{SL}) = Tr\left\{I_{x}\widehat{T}e^{-i\int_{0}^{t_{SL}}dtH_{total}}I_{x}\widehat{T}e^{i\int_{0}^{t_{SL}}dtH_{total}}\right\}, \qquad Eqn. (S2)$$

where  $\hat{T}$  is a Dyson operator. To simplify Eqn. (S1), the total Hamiltonian is transformed into the tilted rf-field frame:1

$$H_{tot}^{rf} = U_{SL}^{-1} H_{tot}(t) U_{SL} - H_{SL}, \qquad Eqn. (S3)$$

where  $U_{SL} = e^{-i\omega_{SL}tI_x}$  is a propagator. The modified Eqn. (S2) in the titled frame is written as follows:

$$S_{SL}(t_{SL}) = Tr\left\{I_x \widehat{T}e^{-i\int_0^{t_{SL}} dt H_{tot}^{rf}} I_x \widehat{T}e^{i\int_0^{t_{SL}} dt H_{tot}^{rf}}\right\}, \qquad Eqn. (S4)$$

since the initial and the measured operators  $(I_x)$  commute with  $U_{SL}$ .

The modified Eqn. (S1) is:

$$H_{tot}^{rf} = 2\pi \sum_{n=1}^{2} a_n \cos(n\omega_R t + \phi_n) \left( I_z \cos(\omega_{SL} t) + I_y \sin(\omega_{SL} t) \right) \widehat{Op}, \qquad Eqn. (S5)$$

where  $\hat{Op}$  remains unchanged as it commutes with  $U_{SL}$ . The Eqn. (S5) can be rewritten in the following way:

$$H_{tot}^{rf} = Eqn. (S6)$$

$$\pi \sum_{n=1}^{2} a_n \left[ \left( \cos\left( (n\omega_R + \omega_{SL})t + \phi_n \right) + \cos\left( (n\omega_R - \omega_{SL})t + \phi_n \right) \right) I_z + \left( \sin\left( (n\omega_R + \omega_{SL})t + \phi_n \right) - \sin\left( (n\omega_R - \omega_{SL})t + \phi_n \right) \right) I_y \right] \widehat{Op}.$$

We see in Eqn. (S6) both  $I_z \widehat{Op}$  and  $I_y \widehat{Op}$  operators, which do not commute with the initial and final operators and are cosine or sine modulated. For small an, these terms can be approximated as zero, except for specific values of the spin lock frequency.

Under specific cases, when  $k\omega_R - \omega_{SL} = 0$  (k=1 or 2), Eqn. (S6) can be rewritten as:

$$H_{tot}^{rf} = \pi a_k [\cos(\phi_n)I_z - \sin(\phi_n)I_y]\widehat{Op} + H_{else}^{rf}, \qquad Eqn. (S7)$$

while  $H_{else}^{rf}$  is:

$$H_{else}^{rf} = \pi \sum_{n=1}^{2} a_n \left[ \cos((n+k)\omega_R t + \phi_n) I_z + \sin((n+k)\omega_R t + \phi_n) I_y \right] \widehat{Op} \qquad Eqn. (S8)$$

 $+a_j[cos((-1)^{\kappa+1}\omega_R t + \phi_n)I_z - sin((-1)^{\kappa+1}\omega_R t + \phi_n)I_y]Op,$ where for the k=1 condition, j=2; and for the k=2 condition j=1. Eqn. (S7) can be further simplified using average Hamiltonian theory,4 considering only the first-order term:

$$T_R H_{tot}^{rf(0)} = \pi T_R a_k \left[ \cos(\phi_n) I_z - \sin(\phi_n) I_y \right] \widehat{Op} = e^{-i\phi_n I_x} \frac{\pi a_k}{\nu_R} I_z \widehat{Op} e^{i\phi_n I_x}, \qquad Eqn. (S9)$$

where the average,  $H_{tot}^{rf(0)}$ , is taken over one rotor period ( $T_R = \frac{1}{\nu_R} = \frac{2\pi}{\omega_R}$ ). Regardless of the explicit form of the  $\widehat{Op}$  operator, the measured spin-lock signal, according to Eqns. (S4) and (S9) is as follows:

$$S_{SL}(t_{SL} = N_{SL}T_R) \approx \cos\left(\pi \frac{a_k}{\nu_R} N_{SL}\right), \qquad Eqn. (S10)$$

since  $e^{-i\phi_n I_x}$  commutes with the initial and final operators. For dipolar interactions, Eqn. (S10) should be modified to account for all orientations:

$$S_{SL}(t_{SL} = N_{SL}T_R) \approx \int d\Omega \cos\left(\pi \frac{a_k}{v_R} N_{SL}\right),$$
 Eqn. (S11)

The integration over orientation ( $\Omega$ ) indicates the powder averaging with Euler angles,  $(\alpha, \beta, \gamma)$ .1

Figure S1 compares numerical (solid lines) and FOH curves (stars) under the rotaryresonance conditions  $v_{SL} = v_R$  (Figures S1A-C) and  $v_{SL} = 2v_R$  (Figures S1D-F) and different  $a_k$  values, related either to  $B_0$  modulation (Figures S1A and D), dipolar interaction (two-spin system, Figure S1B and E) or  $B_1$  modulation (Figures S1C and F). These figures show full agreement between numerical and FOH curves. In all three cases, the changes in  $a_k$  values affect the modulation frequency of the spin-lock signal.

---

## Author Response (AR2)

We thank the Reviewers for their overall positive impression of the revised article. While Reviewer 3 recommended the publication of the article as is, Reviewer 2 suggested a major revision. The reason for a major revision was that our mathematical description involved modulations along  $I_z$ , while physically, these  $B_1$  modulations should only affect the signal when along  $I_y$  (when the spin-lock is along x). Mathematically, there is no difference in the result if the  $B_1$  modulation is along y or z. However, we agree that still we should have highlighted that, physically,  $B_1$  modulation on y contributes, while modulation on z does not. Due to the solenoidal symmetry, no change to the distribution of these modulations is needed, and a relatively minor change in the text addresses this issue. We added the missing part that connects  $B_1$  modulation is with the  $I_z$  operator. This general description is useful since we also discuss  $B_0$ modulations, for which both  $I_z$  and  $I_y$  modulations are relevant.

We also moved the equations to the main text as suggested by the Reviewer.

Additionally, to improve the presentation quality of the article, we have added the following figure to the SI, which visualizes of the inhomogeneity factor and  $B_0$  or  $B_1$  modulations as a 2D color map. We added the following to the SI:

"Figure S8 shows the weighting factors (the color map) according to the range of input inhomogeneity factor values (y-axis) and  $B_0$  modulation amplitude values (x-axis, A) and  $B_1$ modulation amplitude values (x-axis, B). For  $B_0$  modulation (A), the amplitude with the highest value of 200 Hz has the largest weighting factor (red), while for  $B_1$  modulation (B), the opposite is true – the amplitude with 0 value has the largest weighting factor on the total signal. This approximates the expectation that a  $B_0$  field inhomogeneity (e.g. in the y direction) would more strongly affect the sample at the rotor wall, where there is more sample volume. On the other hand, the  $B_1$  field inhomogeneity may be less linear, and affect only a small annulus of sample near the coil. Changes to the distribution mainly affect the location of the first minimum in the SL intensity (see Figure S6 and Figures 4 and 5 in the main text).

**Figure S8** The weighting factor maps are plotted as a function of the inhomogeneity factor (y-axis, in %) and  $B_0(A)$  or  $B_1(B)$  amplitude modulations (x-axis). (A) The amplitudes values range between [0:200] (Hz). (B) The amplitude values range between [0:9.14] (%) with respect to the applied  $v_{SL}$  value. In both color maps, the inhomogeneity factor ranges between [0:5] (%) with respect to the applied  $v_{SL}$  value."

Below is a point-by-point response (in blue) to the second Reviewer's comments (in black).

**Referee #2: Zdeněk Tošner**

(1) The presentation of the pseudo rotary-resonance relaxation dispersion improved with respect to the original submission last year. Including measurements on water sample excludes any doubts this effect is real. Arguments based on average Hamiltonian theory are strong, if assumptions were correct.

I have a fundamental problem with the explanation that the phenomenon is caused by variations of B1 field along z-direction. Typically, rf fields are thought of as perpendicular to the B0 field, that is within the xy plane. In MAS configuration, yes, the coil is tilted at the magic angle and significant rf field is produced along z-axis. BUT!!! This component oscillates at Larmor frequency (it is not affected by transformation into rotating frame) and thus averages out promptly, orders of magnitude faster than MAS.

Thanks for pointing this out. We have modified the presentation to make clear that only for  $B_0$ , a z-modulation survives the secular approximation, while for the  $B_1$  case it does not.

(2) Surprisingly, the importance of z-component modulations is not mentioned in the main text while it is the key component of simulations and theory treatment. It is only vaguely stated on page 10, lines 12-13, "orthogonal to the applied rf field spinlock". Blessed statement! It is not the z-component, it is the y-component, the other orthogonal. Changing direction of Hamiltonian H\_t in equation S1 to I\_y does not influence the conclusions...

We agree with the Reviewer, that changing direction of Hamiltonian Ht in equation S1 from Iz to Iy does not influence the conclusions, and also that the Iy component is the relevant component in relation to B1 modulations. If Ht depends on the Iy operator, we can rotate the total Hamiltonian, Htotal, by  $\frac{\pi}{2}$  around the  $\hat{x}$  axis and obtain the same Hamiltonian as in Eqn. (S1), where it depends on the Iz operator. Since the initial and final operators commute with this rotation, Eqn. (S2) remains exactly the same. Therefore, the Average Hamiltonian Theory (AHT) analysis1 is identical in both cases, whether Ht depends on the Iy or Iz operator.

To clarify, we have modified Eqn. (S1) to make it more general and moved it to the main text, as suggested by the Reviewer.

The added text reads:

"To understand the origin of the pseudo-RRD effect, we start with the simplest case, investigating the behavior of an on-resonance spin (I) during the rf-field spin-lock. The simulated SL-signal is defined as follows:

$$S_{SL}(t_{SL}) = Tr\left\{I_{x}\widehat{T}e^{-i\int_{0}^{t_{SL}}dtH'_{total}}I_{x}\widehat{T}e^{i\int_{0}^{t_{SL}}dtH'_{total}}\right\}, \qquad Eqn. (1)$$

where  $\hat{T}$  is a Dyson operator and  $H'_{total}$  is a total Hamiltonian. We consider the effects of  $B_0$  and  $B_1$  modulations or dipolar interaction. For all three sources,  $H'_{total}$  can be defined as follows:

$$H'_{total} = H'_{SL} + H'_t = \omega_{SL}I_x + Eqn. (2)$$

 $2\pi \sum_{n} a_n \cos(n\omega_R t + \phi_n) [I_z \cos\varphi + I_v \sin\varphi] \widehat{Op},$

where  $\omega_{SL} = 2\pi v_{SL}$  and  $H'_{SL}$  is an ideal spin-lock Hamiltonian. Here,  $\widehat{Op} = 1$  for a single spin with  $B_0 \ (\varphi \ge 0)$  or  $B_1 \ (\varphi = \pi/2)$  modulations, or  $\widehat{Op} = 2S_z$  with  $\varphi = 0$  for a two-spin system (dipolar interaction). While for dipolar interaction, n is 1 or 2,2,3 for  $B_0$  and  $B_1$  modulations, n may take any integer value.4 This is because these modulations are not purely sinusoidal; there are contributions from overtone frequencies. In the experimental SL profiles (Figures 2 and 3), two rotary-resonance conditions are clearly observed. Therefore, in the following discussion, n = 1, 2 will be considered for all three cases. Note also that for the cosine modulated terms of Eqn. 2, only  $I_y$  (and not  $I_z$ ) survives the rotating frame transformation and secular approximation for the case of  $B_1$  modulation. Both terms are relevant for  $B_0$  modulations. For the dipolar interaction,  $a_k$  inversely depend on the distance between the pair of spins and the orientation:2,3  $a_1 = \frac{v_D}{\sqrt{2}} \sin(2\beta)$  and  $a_2 = -\frac{v_D}{2} \sin^2(\beta)$ ;  $v_D = v_{D,IS} = -\frac{\mu_0}{8\pi^2} \frac{\hbar \gamma_I v_S}{r_{IS}^3}$  and ( $\beta$ ) is the Euler angle with respect to the rotor frame.(Mehring, 1983) For  $B_0$  and  $B_1$  modulations,  $a_k$  values do not exhibit any orientation dependence. It is worth noting that for  $B_1$  modulations,  $a_k$  values change with the strength of the applied rf-field lock value ( $v_{SL}$ ).

If  $\varphi$  does not vary with time, Eqn. (2) can be simplified by rotation of  $H'_{total}$  by an  $\varphi$  angle around the  $\hat{x}$  using the operator  $e^{-i\varphi I_x}$ . Such a rotation removes any dependence on  $\varphi$ , since the initial and the final operators in Eqn. (1) commute with  $e^{i\varphi I_x}$ . The modified Eqn. (2) is written as follows:

$$H_{total} = e^{-i\varphi I_x} H'_{total} e^{i\varphi I_x} = H_{SL} + H_t = Eqn. (3)$$

$$\omega_{SL} I_x + 2\pi \sum_n a_n \cos(n\omega_R t + \phi_n) I_z \widehat{Op}.$$

Thus, while  $B_0$  modulation may occur anywhere in the yz-plane, the theoretical treatment remains exactly the same as for z modulation. Mathematically, this is also true for  $B_1$  modulation, while physically, these modulations are only relevant when in the transverse plane."

(3) Average Hamiltonian treatment is a convincing proof of the phenomenon and should be moved to the main text. At least Equation S1 highlighting variations along z-direction for B0, and along ydirection for B1. And the result, like Eq. S10, but with better explanation what is the frequency of signal amplitude modulations and what is a time (in Eq. S10, N\_SL supplements the time axis, but it depends on MAS frequency and thus is confusing). It should be clear that the frequency of signal amplitude modulations does not depend on MAS in case of B0 variations, but it does depend on MAS in case of B1 variations (it is clear from simulations in the figure S4 but it is possible to write a formula as well)

We now clarify that the z-direction (and y) is relevant for  $B_0$  and the y-direction is relevant for  $B_1$ . We added the sentence regarding dependence of the frequency of signal amplitude modulations for  $B_0$  and  $B_1$ :

"In the case of  $B_0$  modulation, adjustments to the shimming coil are expected to have a profound effect, but oscillations in signal amplitude are expected to be independent of the applied B1 field. By contrast, for  $B_1$  modulation, changes in the strength of the applied spin-lock have a major effect, since the oscillation frequency of signal amplitude is expected to depend on  $B_1$ ."

(4) I suggest major revisions. It must be highlighted and evident from the first reading of the abstract and the paper which component of rf field is causing the phenomenon.

While mathematically there is no difference if  $B_1$  modulation occurs along  $I_y$  or  $I_z$  or both at the same time, physically, only  $I_y$  is relevant (when the spinlock is on x).

(5) Just a note to the author's response to Reviewer #1, point 2, regarding rf-field inhomogeneity profile. The figure included in the response documents common misunderstanding in the community. The nutation profile is NOT symmetric, and it has an important tail towards lower frequencies which is easily overlooked in data suffering from truncation artifacts like those shown in the figure. Best analytical function to fit nutation profiles is not a gaussian but the power law suggested by Gupta et al. 2015. Gaussian captures most abundant features though...

We added the following sentences to the main text:

"A more quantitative assessment would call for calculation of the exact values and shapes of  $B_1$  fields for a particular coil, as well as better characterization of  $B_0$  distribution.4–14 Note that the magnitudes within these distributions are reasonable, considering the published calculations for solenoidal coils.7,15,16"

- Haeberlen, U.; Waugh, J. S. Coherent Averaging Effects in Magnetic Resonance. *Phys. Rev.* 1968, 175 (2), 453–467. https://doi.org/10.1103/PhysRev.175.453.
- (2) Mehring, M. *Principles of High Resolution NMR in Solids*, 2nd ed.; Springer-Verlag: Berlin Heidelberg, 1983. https://doi.org/10.1007/978-3-642-68756-3.
- (3) Olejniczak, E. T.; Vega, S.; Griffin, R. G. Multiple Pulse NMR in Rotating Solids. J. Chem. *Phys.* **1984**, *81* (11), 4804–4817. https://doi.org/10.1063/1.447506.
- (4) Aebischer, K.; Tošner, Z.; Ernst, M. Effects of Radial Radio-Frequency Field Inhomogeneity on MAS Solid-State NMR Experiments. *Magn. Reson.* **2021**, *2* (1), 523–543. https://doi.org/10.5194/mr-2-523-2021.
- (5) Hürlimann, M. D.; Griffin, D. D. Spin Dynamics of Carr–Purcell–Meiboom–Gill-like Sequences in Grossly Inhomogeneous *B*0 and *B*1 Fields and Application to NMR Well Logging. *J. Magn. Reson.* 2000, *143* (1), 120–135. https://doi.org/10.1006/jmre.1999.1967.
- (6) Paulson, E. K.; Martin, R. W.; Zilm, K. W. Cross Polarization, Radio Frequency Field Homogeneity, and Circuit Balancing in High Field Solid State NMR Probes. *J. Magn. Reson.* 2004, 171 (2), 314–323. https://doi.org/10.1016/j.jmr.2004.09.009.
- (7) Tošner, Z.; Purea, A.; Struppe, J. O.; Wegner, S.; Engelke, F.; Glaser, S. J.; Reif, B. Radiofrequency Fields in MAS Solid State NMR Probes. *J. Magn. Reson.* 2017, 284, 20–32. https://doi.org/10.1016/j.jmr.2017.09.002.
- (8) Tošner, Z.; Sarkar, R.; Becker-Baldus, J.; Glaubitz, C.; Wegner, S.; Engelke, F.; Glaser, S. J.; Reif, B. Overcoming Volume Selectivity of Dipolar Recoupling in Biological Solid-State NMR Spectroscopy. *Angew. Chem. Int. Ed.* **2018**, *57* (44), 14514–14518. https://doi.org/10.1002/anie.201805002.
- (9) Guenneugues, M.; Berthault, P.; Desvaux, H. A Method for Determining*B*1Field Inhomogeneity. Are the Biases Assumed in Heteronuclear Relaxation Experiments Usually Underestimated? *J. Magn. Reson.* **1999**, *136* (1), 118–126. https://doi.org/10.1006/jmre.1998.1590.
- (10) Odedra, S.; Wimperis, S. Imaging of the *B*1 Distribution and Background Signal in a MAS NMR Probehead Using Inhomogeneous *B*0 and *B*1 Fields. *J. Magn. Reson.* 2013, 231, 95–99. https://doi.org/10.1016/j.jmr.2013.04.002.
- (11) Engelke, F. Electromagnetic Wave Compression and Radio Frequency Homogeneity in NMR Solenoidal Coils: Computational Approach. *Concepts Magn. Reson.* 2002, *15* (2), 129–155. https://doi.org/10.1002/cmr.10029.

- (12) Privalov, A. F.; Dvinskikh, S. V.; Vieth, H.-M. Coil Design for Large-Volume High-B1Homogeneity for Solid-State NMR Applications. J. Magn. Reson. A 1996, 123 (2), 157– 160. https://doi.org/10.1006/jmra.1996.0229.
- (13) Lips, O.; Privalov, A. F.; Dvinskikh, S. V.; Fujara, F. Magnet Design with High B0 Homogeneity for Fast-Field-Cycling NMR Applications. J. Magn. Reson. 2001, 149 (1), 22– 28. https://doi.org/10.1006/jmre.2000.2279.
- (14) Schönzart, J.; Han, R.; Gennett, T.; Rienstra, C. M.; Stringer, J. A. Magnetic Susceptibility Modeling of Magic-Angle Spinning Modules for Part Per Billion Scale Field Homogeneity. J. Magn. Reson. 2024, 364, 107704. https://doi.org/10.1016/j.jmr.2024.107704.
- (15) Gupta, R.; Hou, G.; Polenova, T.; Vega, A. J. RF Inhomogeneity and How It Control CPMAS. *Solid State Nucl. Magn. Reson.* 2015, 72, 17–26. https://doi.org/10.1016/j.ssnmr.2015.09.005.
- (16) Uribe, J. L.; Jimenez, M. D.; Kelz, J. I.; Liang, J.; Martin, R. W. Automated Test Apparatus for Bench-Testing the Magnetic Field Homogeneity of NMR Transceiver Coils. J. Magn. Reson. Open 2024, 18, 100142. https://doi.org/10.1016/j.jmro.2023.100142.